# A distal enhancer guides the negative selection of toxic glycoalkaloids during tomato domestication

Feng Bai[1,7], Peng Shu[1,2,7], Heng Deng[1], Yi Wu[1], Yao Chen [1], Mengbo Wu[1], Tao Ma [1], Yang Zhang [1], Julien Pirrello[3], Zhengguo Li [4], Yiguo Hong [5,6], Mondher Bouzayen[1,3] ✉ & Mingchun Liu [1] ✉

Steroidal glycoalkaloids (SGAs) are major plant defense metabolites against pests, while they are considered poisonous in food. The genetic basis that guides negative selection of SGAs production during tomato domestication remains poorly understood. Here, we identify a distal enhancer, *GAME Enhancer 1* (*GE1*), as the key regulator of SGAs metabolism in tomato. *GE1* recruits MYC2-GAME9 transcriptional complex to regulate the expression of *GAME* cluster genes via the formation of chromatin loops located in the neighboring DNA region. A naturally occurring *GE1*[76] allelic variant is found to be more active in stimulating *GAME* expression. We show that the weaker *GE1* allele has been the main driver for selecting reduced SGAs levels during tomato domestication. Unravelling the "TFs-Enhancer-Promoter" regulatory mechanism operating in SGAs metabolism opens unprecedented prospects for SGAs manipulation in *Solanaceae* via precision breeding strategies.

*Solanaceae* crops have taken a preponderant part in the human diet over the past centuries and one needs only think of the great famine caused by the potato disease that led to more than a million deaths in the 19th century. Domestication of many *Solanaceae* for human foods and vegetables has faced the prior challenge of ridding them of their toxic steroidal glycoalkaloids (SGAs) compounds[1]. SGAs are specialized secondary metabolites produced in *Solanaceae* crops, such as tomato (*Solanum lycopersicum*), potato (*Solanum tuberosum*), and eggplant (*Solanum melongena*)[1–3]. They are important compounds for plant defense against a variety of pathogens and herbivores but undesirable in food and feed due to their anti-nutritional and toxic activities[3,4]. In tomato and potato, SGAs biosynthesis pathway starts from cholesterol and undergoes a series of hydroxylation, acetylation, and glycosylation steps mediated by a set of *GLYCOALKALOID*

*METABOLISM* (*GAME*) genes which are physically clustered in the genomes[2,3,5–7].

GAME9, an AP2/ERF transcription factor (TF), acts as key regulator of SGAs biosynthesis in tomato through its binding to promoters of core *GAME* genes[8,9]. Notably, the variant allele *GAME9*[135V], selected during tomato domestication and present in big-fruited modern varieties, displays weak DNA binding capacity and cooperates with other intermediate TFs, such as MYC2, to regulate the expression of *GAME* genes[8–10]. Strikingly, neither GAME9 nor MYC2 is able to directly bind the promoters of *GAME1* and *GAME18* genes, yet described as being regulated by GAME9-MYC2 complex genes[9,11]. This raises the hypothesis that the two TFs modulate the transcriptional level of *GAME* genes through a yet unknown regulatory mechanism. Given that this metabolic gene clusters are confined within a topologically associating

[1]Key Laboratory of Bio-Resource and Eco-Environment of Ministry of Education, College of Life Sciences, Sichuan University, Chengdu 610065 Sichuan, China. [2]Clinical Medical Research Center, Xinqiao Hospital, Army Medical University, Chongqing 400037, China. [3]Laboratoire de Recherche en Sciences Végétales-Génomique et Biotechnologie des Fruits-UMR5546, Université de Toulouse, CNRS, UPS, Toulouse-INP, Toulouse, France. [4]Key Laboratory of Plant Hormones and Development Regulation of Chongqing, School of Life Sciences, Chongqing University, Chongqing, China. [5]School of Life Sciences, University of Warwick, Warwick CV4 7AL, UK. [6]State Key Laboratory of North China Crop Improvement and Regulation and College of Horticulture, Hebei Agricultural University, Baoding 071000, China. [7]These authors contributed equally: Feng Bai, Peng Shu. ✉e-mail: mondher.bouzayen@toulouse-inp.fr; mcliu@scu.edu.cn

domain (TAD)[12,13], we postulated that enhancers might be involved in their regulation.

Enhancers are distal cis-regulatory DNA sequences known as core regulators governing spatiotemporally and quantitatively the expression of target genes in an orientation-, position- and distance-independent manner[14–16]. With increasing chromatin accessibility and dynamic histone modifications, enhancers can be activated and brought into the proximity of their target gene promoters to form enhancer-promoter chromatin loops[12,17,18]. Through recruiting TFs and co-factors, they can physically interact with target gene promoters and then trigger their expression[18–22]. Although *GAME* genes represent a well-known metabolic gene cluster, so far, any putative involvement of enhancers in controlling their expression remained unexplored.

We report here on the characterization of a tomato T-DNA insertional mutant displaying wrinkled fruit surface (*ws*) that guided the identification of *GAME enhancer 1* (*GE1*) topologically associating with *GAME* cluster genes region in chromosome 7. We demonstrate that *GE1* recruits MYC2-GAME9 transcriptional complex to physically bind the promoter of *GAME* genes through an enhancer-promoter chromatin loop leading to the upregulation of the *GAME* cluster genes. The T-DNA insertion disrupted the "TFs-enhancer-promoter" regulatory module resulting in the wrinkled surface phenotype of tomato fruits due to excessive accumulation of γ-tomatine. Intriguingly, we also identified a naturally occurring *GE1*[76] allelic variant in domesticated tomato accessions, showing that *GE1* rather than *GE1*[76] has been selected for reduced SGAs content during tomato domestication.

## Results

### The wrinkled surface (*ws*) mutant is induced by a T-DNA insertion in a functional locus

In an attempt to overexpress a MADS-box gene (*AGL66*, Solyc07g052700) in the MicroTom tomato cultivar, we identified a single mutation event that resulted in wrinkled surface fruit in transgenic line *AGL66*-OE8 (*ws* mutant) (Fig. 1a and Supplementary Fig. 1a). The *ws* mutant displayed irregular and abnormal epidermal cells, uneven and concave surface and morphologically altered and chaotically positioned exocarp cells (Fig. 1b and Supplementary Fig. 1b, c). Strikingly, such *ws* phenotypic changes were not correlated with overexpression, or RNAi-mediated downregulation of the MADS-box transgene, but were associated with the transgene presence in the transgenic plants (Supplementary Fig. 1a, d, e). Moreover, genetic cross 'WT (♂) x *ws* (♀)' showed a segregation ratio of 1:3 between the wrinkled vs smooth surface fruits in the $F_2$ generation, and only homozygous mutant plants produced wrinkled surface fruits (Fig. 1c and Supplementary Fig. 1f, g). We interpret these data to mean that the *ws* mutation might be due to a random T-DNA insertion in a locus responsible for the defective phenotypes.

To investigate the T-DNA insertion site, we performed whole-genome sequencing of the *ws* mutant uncovering a single copy T-DNA insertion located into the intergenic region 57,125,923 to 57,125,938 of chromosome 7 (chr07) (Fig. 1d and Supplementary Data 1). However, no protein-coding gene was found within approximately 10-kb sequences around the T-DNA insertion site. The closest genes are those encoding glycosyltransferase, namely *GAME17* located at 11,481-bp upstream, and *GAME1* and *GAME18* located at 33,292-bp and 52,059-bp downstream of the T-DNA insertion site (Fig. 1d), respectively.

### SGAs metabolism altered in *ws* mutant

Because *ws* fruits phenocopy the woody aspect with suberized regions around the fruit apex of *GAME1*-RNAi lines due to tomatidine accumulation[2], and given that the presence of three *GAME* genes localizing close to the T-DNA site (Fig. 1d), we investigated the impact of the *ws* mutation on the expression of these genes by RNA-seq profiling of pericarp tissue in WT and *ws* fruits at 7 DPA (Supplementary Data 2). KEGG and GO analyses pointed to a dysregulation of SGAs

metabolism genes which may had led to abnormal accumulation of specific metabolites resulting in altered development of epidermal cells and wrinkled surface of *ws* fruits (Supplementary Fig. 2a, b and Supplementary Data 3, 4). Close examination of the transcript levels of SGAs metabolic genes revealed a significant increase in the expression of genes acting upstream of γ-tomatine, whilst those acting downstream of γ-tomatine exhibited reduced expression levels in *ws* compared to WT fruits (Supplementary Fig. 2c). Transcript accumulation assessed by RT-qPCR was consistent with the RNA-seq data for the six *GAME* genes (Supplementary Fig. 2d), suggesting that the SGAs biosynthesis pathway was affected in a way that led to γ-tomatine accumulation in the *ws* mutant.

To investigate whether the wrinkled surface phenotype of *ws* fruits is caused by the accumulation of phytotoxic SGAs compounds, we assessed SGAs content in fruit extracts by UPLC-MS revealing highly significant changes in γ-tomatine compared to other metabolites (Fig. 1e, Supplementary Fig. 3a–f; Supplementary Data 5). Compared to WT, γ-tomatine content increased in *ws* mutants at different fruit developmental stages reaching 16.1-fold and 20.4-fold higher levels at 7- and 17-DPA, respectively (Supplementary Fig. 3d). The dramatic increase in γ-tomatine was consistent with the upregulation of *GAME1* and *GAME17*, and the down-regulation of *GAME18* in *ws* mutant fruits. These findings suggest that the T-DNA insertion affects the GAME cluster genes involved in SGAs metabolism, and that the wrinkled surface phenotype is most likely caused by the accumulation of phytotoxic γ-tomatine due to the aberrant expression of *GAME* genes in the *ws* mutant (Fig. 1f).

The data described above support the idea that the wrinkled surface in the *ws* mutant is caused by the downregulation of *GAME18* induced by the T-DNA insertion. To validate this hypothesis, we generated *GAME18*-knockout tomato lines (*game18*) by CRISPR/Cas9 system. The *game18* mutants generated produce a truncated *GAME18* protein that is 43-amino acid long anticipated to be non-functional (Fig. 1g). Interestingly, *game18* mutants produce wrinkled fruit surfaces similar to the phenotypes exhibited by *ws* mutants (Fig. 1h). Assessing SGAs content in *game18* fruits by UPLC-MS indicated that γ-tomatine levels were increased (Fig. 1i), in contrast to those of tomatidenol, tomatidine, tomatidine galactoside, β1-tomatine and α-tomatine that were all significantly decreased (Supplementary Fig. 3g). While confirming the role of *GAME18* in SGAs metabolism, these data are consistent with the idea that the wrinkled surface of *ws* fruits is due to the downregulation of *GAME18* which leads to high accumulation of γ-tomatine.

### GE1 acts as an active enhancer

In the absence of any gene in the close vicinity of the T-DNA insertion site, we assumed that this intergenic region may contain sequences with potential regulatory function. Mining available data for DNase I hypersensitivity site (DHS) and histone modification profiles[23,24], we identified a region of 2.9-kb near the T-DNA insertion site that shifts from high to low levels of chromatin accessibility and H3K27ac and H3K4me3 histone marks from 7 to 17 DPA fruits, while by contrast, H3K27me3 histone mark showed the opposite trend (Fig. 2a). Interestingly, DHS and histone modifications in the promoter regions of *GAME17*, *GAME1* and *GAME18* surrounding the T-DNA insertion site did not display such dynamic changes (Supplementary Fig. 4a–d). The DHS and histone modification features raise the hypothesis that the 2.9-kb region may function as an enhancer of *GAME* genes. To test this hypothesis, we performed ChIP-qPCR to determine the RNA polymerase II (RNAPII) signals and H3K27ac level within the 2.9-kb region considering that active enhancers are usually enriched in H3K27ac histone marks and recruit RNAPII. The data show that RNAPII signals and histone H3K27ac marks are highly enriched within the five fragments P1-P5 of the 2.9-kb region in fruit tissues at 7- and 17-DPA (Fig. 2b–e). Moreover, both RNAPII signals and H3K27ac modification levels significantly decreased

within P3-P5 fragments in *ws* vs WT fruits at 7-DPA (Fig. 2b, d). Based on the typical enhancer features exhibited by this 2.9-kb region, we therefore designated it as *GAME Enhancer 1* (*GE1*).

To test which part of *GE1* functions as the key fragment of enhancer, we segmented the 2.9-kb fragment into three sub-fragments (EI, EII and EIII) based on the presence of typical enhancer features (Fig. 2a). The three sub-fragments were subsequently used in trans-activation assays following cloning in the pGL3 vector harboring the firefly luciferase (*LUC*) reporter gene fused to a minimal *35S* promoter (Supplementary Fig. 4e). The data indicated that the entire *GE1* fragment promotes higher transcriptional activity than EI, EII, or EIII sub-

fragments, emphasizing the need for all three sub-fragments to have a fully functional *GE1* (Fig. 2f). Given that enhancer RNAs (eRNAs) are known to be transcribed from active enhancers and are associated to enhancer-promoter loops in regulating transcription[25], we then performed PCR assays on WT and *ws* fruit tissues showing that *GE1* RNA transcripts were successfully detected in WT and *ws* fruits at 7 DPA (Supplementary Fig. 4f). Moreover, RT-qPCR assays of relative transcript levels showed higher accumulation of the truncated EI, EII and EIII transcripts in *ws* than in WT (Fig. 2g). Taken together, these data indicate that the 2.9-kb *GE1* displays the necessary features to work as an active enhancer.

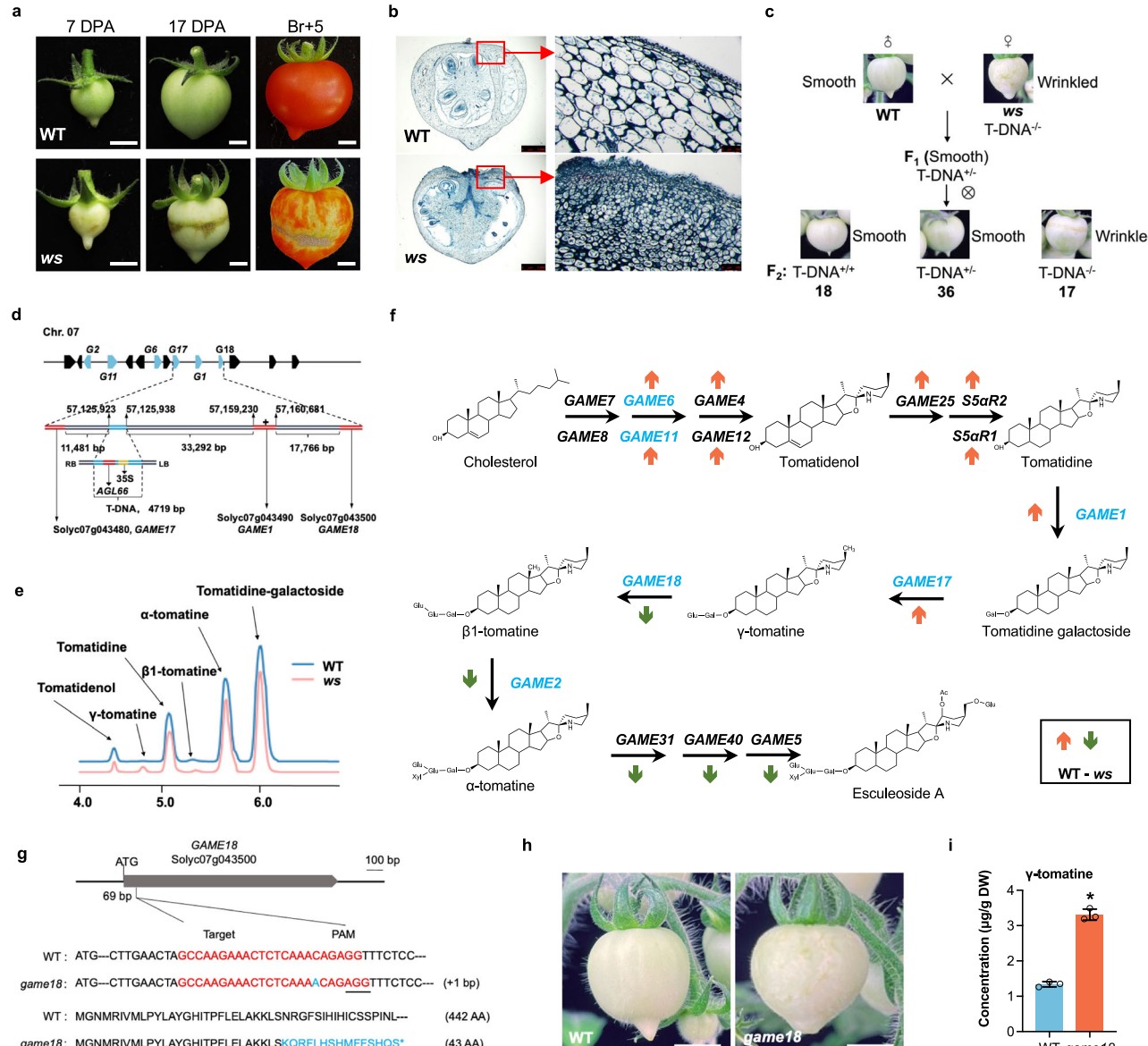

**Fig. 1 | The *ws* tomato mutant shows alteration in SGAs metabolism. a** Different ripening stages in WT and *ws* mutant. DPA, Days post anthesis; Br+5, Breaker plus 5 days. The scale bar represents 5 mm (7 DPA) and 10 mm (17 DPA and Br+5). **b** Microscopic cross-sections of 17 DPA fruits stained with safranin-fixed green. Photographed were taken under a stereoscopic microscope. Bar = 2.5 mm (left), and 100 μm (right). The picture is representative of at least 5 independent fruits of mutant all showing similar results. **c** Genetic crosses. The *ws* mutant was crossed to the WT resulting in smooth surface F₁ hybrid. F₁ progeny was self-fertilized to generate F₂ progeny. **d** Schematic representation of the T-DNA insertion in the intergenic region between six clustered *GAME* genes on tomato chr07. The six *GAME* genes are indicated in blue. *G* stand for *GAME*. **e** SGAs profile of 7 DPA fruits

of WT and *ws*. **f** Expression of structural genes in the SGAs biosynthesis pathway. The *GAME* genes clustered on chr07 are shown in blue. The direction of the arrows indicates up- (highlighted in brown) or down- regulation (highlighted in green) of gene expression in *ws* compared to WT. The direction of the arrows indicates up- (highlighted in brown) or down- regulation (highlighted in green) of gene expression in *ws* compared to WT. **g** Generation of *GAME18* knockout lines (*game18*) in MicroTom by CRISPR/Cas9. The gRNA is highlighted red, and PAM sequence is underlined. Mutations within *GAME18* coding sequences are shown in blue. AA, amino acids. **h** The phenotypes of WT and *game18* mutant fruits at 17 DPA. Bar = 1 cm. **i** Concentration of γ-tomatine in WT and *game18* fruit at 17 DPA. The value is presented as the mean ± SD (*n* = 3). *P < 0.05 in two-sided *t* test.

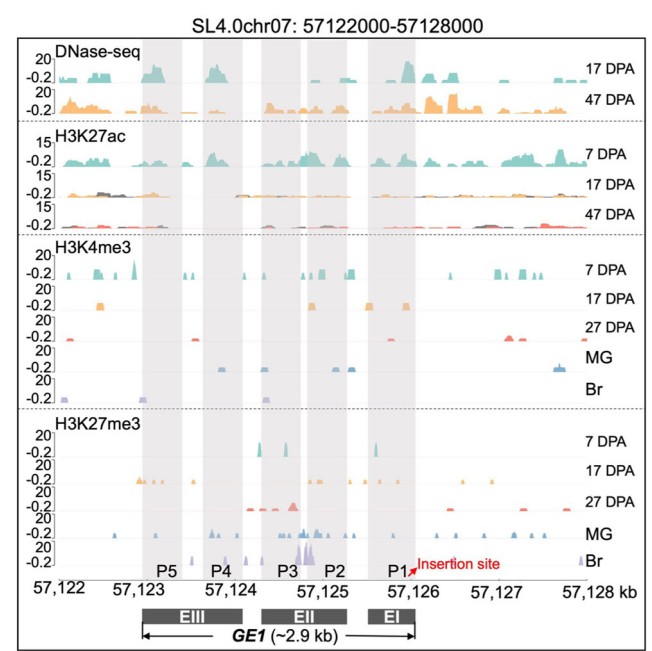

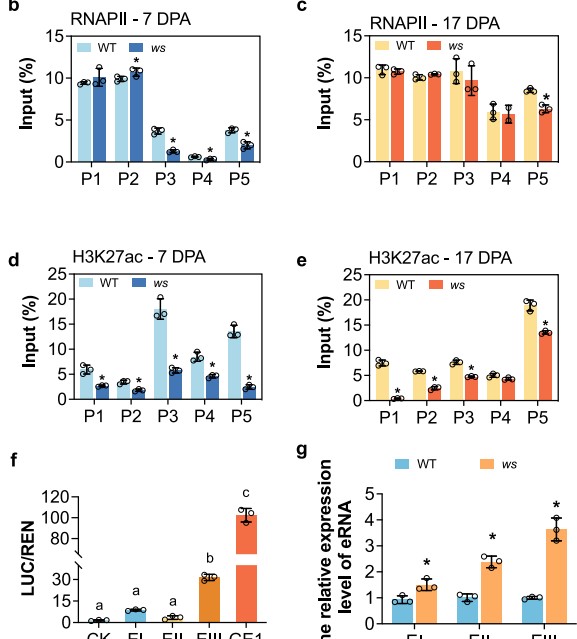

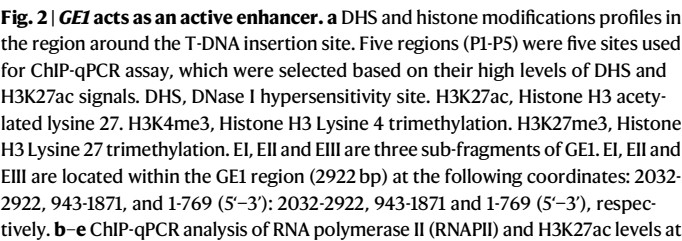

**Fig. 2 | GE1 acts as an active enhancer. a** DHS and histone modifications profiles in the region around the T-DNA insertion site. Five regions (P1-P5) were five sites used for ChIP-qPCR assay, which were selected based on their high levels of DHS and H3K27ac signals. DHS, DNase I hypersensitivity site. H3K27ac, Histone H3 acetylated lysine 27. H3K4me3, Histone H3 Lysine 4 trimethylation. H3K27me3, Histone H3 Lysine 27 trimethylation. EI, EII and EIII are three sub-fragments of GE1. EI, EII and EIII are located within the GE1 region (2922 bp) at the following coordinates: 2032-2922, 943-1871, and 1-769 (5′-3′): 2032-2922, 943-1871 and 1-769 (5′-3′), respectively. **b**–**e** ChIP-qPCR analysis of RNA polymerase II (RNAPII) and H3K27ac levels at P1-P5 of GE1 region in WT and ws fruits at 7 DPA and 17 DPA. Data are presented as mean ± SD ($n = 3$). *$P < 0.05$ in two-sided $t$ test. **f** Transient assays showing the enhancer activity of EI, EII, EIII and GE1 on gene expression. Different letters above the bars represent significant differences. Data are presented as mean ± SD ($n = 3$). $P < 0.05$ in one-way ANOVA. CK, the empty pGL3 vector. **g** RT-qPCR assays of relative transcript levels of three fragments from GE1 in WT and ws fruits. Data are presented as mean ± SD ($n = 3$). *$P < 0.05$ in multiple unpaired tests using Benjamini–Hochberg methods.

## GE1 modulates GAME gene cluster by forming chromatin loops

To address whether the T-DNA insertion alters the chromatin looping usually associated with functional enhancers, we performed high-throughput chromosome conformation capture (Hi-C) sequencing (Supplementary Data 6). This approach provided a panorama view of chromatin architectures at 7-DPA tomato fruits allowing to analyze the genome organization at the chromosome level with combined Hi-C maps at different resolutions (Fig. 3a–f). The data revealed the co-regulation of the GAME genes cluster, including GAME2, GAME11, GAME6, GAME17, GAME1, and GAME18, located within a topologically associating domain (TAD) spanning 56.9–57.2 Mb on chr07 (Fig. 3c, f). Notably, GE1 and the three GAME cluster genes (GAME1, GAME17, and GAME18) which exhibit a similar expression pattern during fruit development (Supplementary Fig. 5a), are associated in a large chromatin loop composed of three loops spanning the region 57.1–57.2 Mb in WT (Fig. 3c, g). Interestingly, these chromatin loops are totally lacking in ws mutant fruit (Fig. 3f, h). These findings suggest that the T-DNA insertion in ws mutant hampers the formation of the chromatin loops, thus impairing the GE1-mediated regulation of GAME genes cluster.

Investigation of the regulation of the three GAME genes by GE1 using transactivation assays (Fig. 3i) revealed that the activity of LUC driven by GE1-proG17, GE1-proG1 and GE1-proG18 increased 4.5-fold, 6.8-fold, and 17.9-fold, respectively, compared the control assay (Fig. 3j). Interestingly, $GE1_{T-DNA}$ fragment, constructed to mimic GE1 with the T-DNA insertion in ws mutant, displayed significantly reduced activation activity compared to GE1 (Fig. 3k). Collectively, our data are consistent with the notion that GE1 spatially interacts with the GAME cluster genes via formation of chromatin loops, which plays a critical role in modulating the expression of GAME cluster genes.

To further address the GE1 enhancer function, we implemented a genetic approach consisting of generating three GE1-knockout tomato lines (ge1-1, ge1-2, and ge1-3) by CRISPR/Cas9 system. PCR analysis showed that all three ge1 lines had a large deletion (>1.6 kb) that was confirmed by sequencing (Fig. 4a). Importantly, all three ge1 lines displayed wrinkled surfaces that phenocopy the ws mutants (Fig. 4b). Moreover, transcript levels of GAME1, GAME17 and GAME18, shown to be directly regulated by GE1, were significantly lower in all three ge1 lines than in WT (Fig. 4c). Consistent with the expression levels of GAME genes, the content of tomatidenol and tomatidine, two toxic SGAs compounds, were significantly increased, while that of tomatidine galactoside, γ-tomatine, β1-tomatine and α-tomatine were decreased in the three ge1 compared to WT (Fig. 4d). Taken together, these data provide direct genetic evidence supporting the requirement of a functional GE1 in the regulation of the GAME gene cluster.

## A natural GE1 mutation has been selected during tomato breeding

During tomato domestication going from S. pimpinellifolium (PIM) to S. lycopersicum var. cerasiforme (CER) and to S. lycopersicum (BIG), SGAs have been negatively selected with the aim to reduce toxicity and fruit bitterness. Interestingly, analysis of 46 tomato genomes including the PIM, CER and BIG tomato groups (Supplementary Data 7), uncovered an allelic variant of GE1. This allele showed a 76 bp deletion during domestication, hence named it as $GE1^{76}$ (Fig. 5a and Supplementary Data 8). Remarkably, the frequency of the $GE1^{76}$ variant allele decreased from 54.5% in PIM, to 18.1% in CER, and to 8.3% in BIG tomato (Fig. 5b). Moreover, publicly available metabolic data[10] show that the content of five main SGAs in 22 accessions (4 $GE1^{76}$ and 18 GE1) was

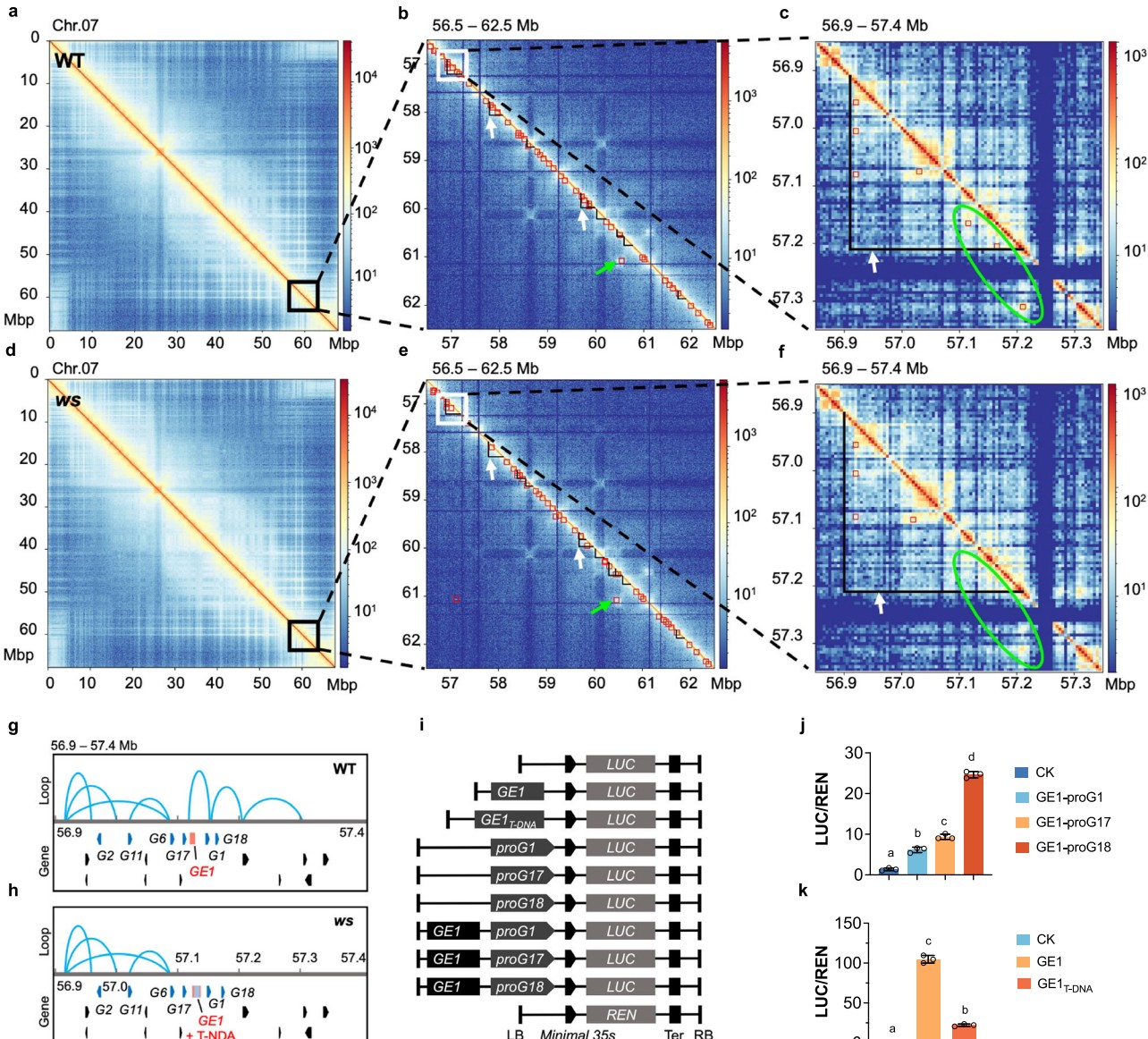

**Fig. 3 | Hi-C based high-resolution 3D genome mapping in tomato fruits.**
Chromatin interaction heatmaps of chromosome 7 in WT and *ws* fruits at the
100 kb (**a**, **d**),10 kb (**b**, **e**), or 5 kb (**c**, **f**) resolution. The solid black line represents the topologically associating domains (TADs), pointed out with white arrows. The red box represents the chromatin interactions region, pointed out with green arrows. The difference of chromatin interactions region was marked with green circles. The interactions between *GE1* region and the three GAME cluster genes (*GAME1*, *GAME17*, and *GAME18*) in WT are the two red boxes in the top of the green circles, and these interactions disappeared in *ws* mutant because of the T-DNA insertion. **g**, **h** Chromatin loops between *GE1* and *GAME* genes in WT and *ws* fruits are plotted based on Hi-C data sets of (**c**) and (**f**), respectively. Chromatin interactions identified by Hi-C are represented by blue lines. *G1*, *G2*, *G6*, *G11*, *G17* and *G18* stand for *GAME1*, *GAME2*, *GAME6*, *GAME11*, *GAME17* and *GAME18*, respectively. **i** Schematic illustration of the reporter and effector plasmid used in the transient assay. LB and RB, the left and right boundary of T-DNA. *LUC*, the ORF of *luciferase*. *REN*, the ORF of *Renilla luciferase*. TER, nopaline synthase terminator. **j** Transient assays showing the enhancer activity of *GE1* on the promoter of *GAME1* (*proG1*), *GAME17* (*proG17*), and *GAME18* (*proG18*). Data are presented as mean ± SD (*n* = 3). There are significant differences between the different letters (i.e., **a**, **b** etc) above the bars. *P* < 0.05 in one-way ANOVA. **k** Transient assays showing the enhancer activity of *GE1* and *GE1*$_{T-DNA}$. Data are presented as mean ± SD (*n* = 3). There are significant differences between the different letters (i.e., **a**, **b** etc) above the bars. *P* < 0.05 in one-way ANOVA.

reduced in *GE1*-containing accessions compared to the accession containing the *GE1$^{76}$* allele (Fig. 5c and Supplementary Data 9). Given the low frequency of the *GE1$^{76}$* variant in the BIG modern tomato, we assume that selection operated against this allele variant during tomato domestication as to reduce toxic SGAs content.

As an additional line of evidence supporting the active role of *GE1* in modulating SGAs profiles, RT-qPCR analysis and SGAs profiling were performed in 19 tomato varieties including 13 *GE1* accessions (TS2, TS3, TS9, TS15, TS60, TS96, TS118, TS166, TS171, TS185, TS204, TS256, and TS439) and 6 *GE1$^{76}$* accessions (TS22, TS39, TS117, TS156, TS413,

and LA2093). The transcript levels corresponding to the clustered *GAME* genes in chr07 were significantly higher in *GE1$^{76}$* accessions than in *GE1* accessions (Fig. 5d). In line with the higher expression levels of *GAME* genes, the content of the main SGAs was also higher in *GE1$^{76}$* than in *GE1* accessions (Fig. 5e). Consistently, *GE1$^{76}$* displayed higher enhancer activity than *GE1* (Fig. 5f and Supplementary Fig. 5b). These data sustain the idea that *GE1* was an important driver in the process of reducing SGAs content during tomato domestication and that the weaker allele *GE1*, rather than *GE1$^{76}$*, has been preferentially selected in modern tomato cultivars.

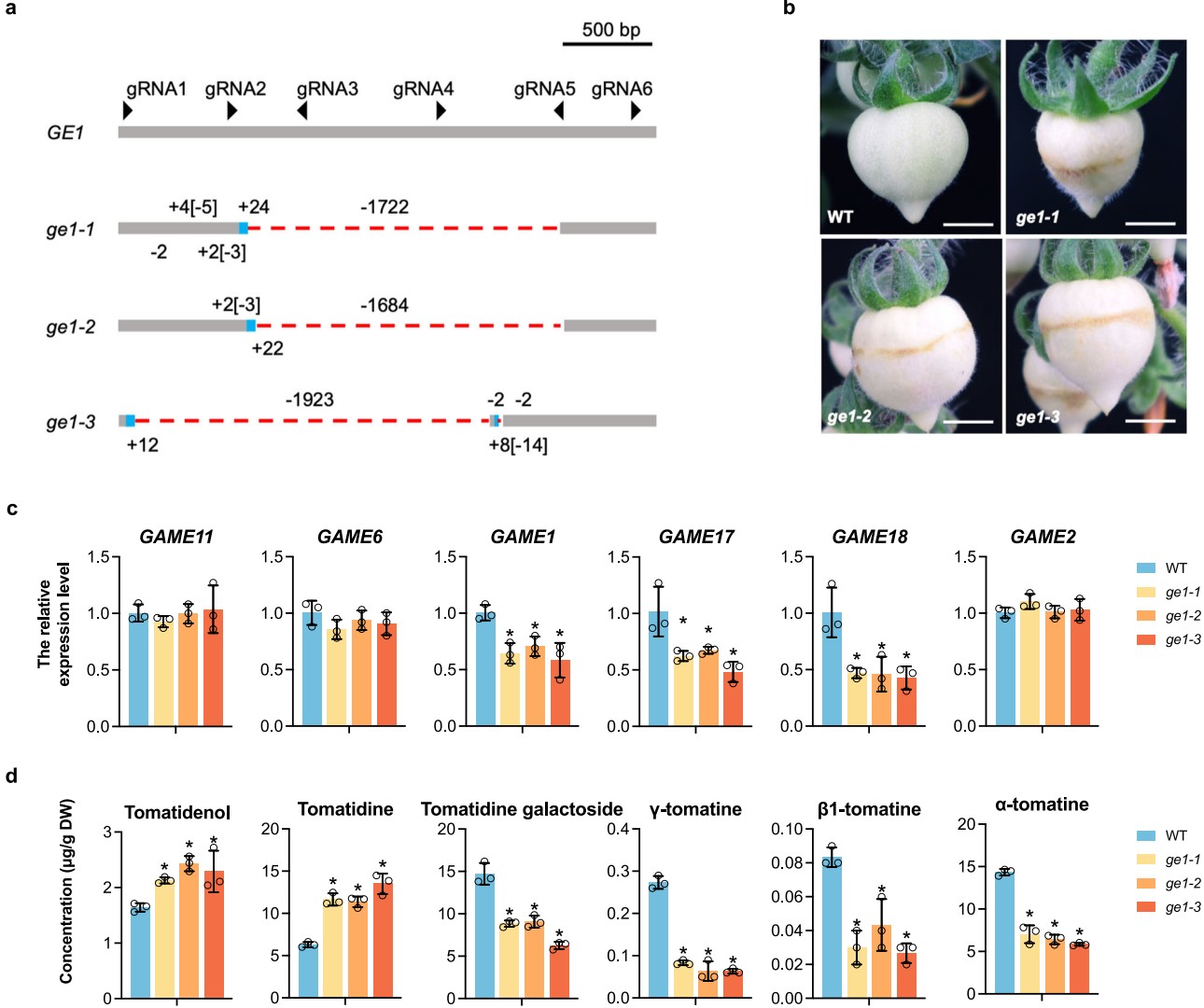

**Fig. 4 | Knockout of *GE1* results in a wrinkled surface of fruit. a** Generation of *GE1* knockout lines in MicroTom tomato by CRISPR/Cas9 mediated genome editing technology. *ge1-1*, *ge1-2*, and *ge1-3* are three *GE1* knockout lines. Deletions (−) and insertions (+) indicated by numbers. Different gRNAs are presented by black arrowheads. **b** The phenotypes of three *ge1* mutant fruits at 17 DPA. Bar = 1 cm. **c** RT- qPCR analysis of the relative expression of six *GAME* genes in WT and three *ge1* lines at 17 DPA. Data are presented as mean ± SD (*n* = 3). *P < 0.05 in two-sided *t* test. **d** SGAs profiles of WT and *ge1* fruits at 17 DPA. UPLC-MS was used for SGAs profiling. Data are presented as mean ± SD (*n* = 3). *P < 0.05 in two-sided *t* test.

## GE1 recruits MYC2-GAME9 TF complex

Previous studies have showed that enhancers regulate expression of target genes by recruiting specific TFs and their associated co-factors[14,15]. In light of the data presented above showing that *GAME* cluster genes are the main target of *GE1*, we mined our RNA-seq data to extract the DEGs encoding TF known to be involved in the regulation of SGAs biosynthesis. Among the SGAs regulator genes MYC1, MYC2, GAME9 and MYB12 exhibited increased expression in *ws* mutant (Supplementary Fig. 6a; Supplementary Data 10). Notably, MYC2 and GAME9 have been previously shown to form a transcriptional complex that regulates the expression of SGAs biosynthesis genes[8,9]. Moreover, although neither MYC2 nor GAME9 can directly bind to the *GAME18* promoter, *GAME18* transcript levels were altered in both MYC2 and GAME9 transgenic lines[9,11]. Given that *GE1* displays its highest activation capacity on the *GAME18* promoter compared to other *GAME* cluster genes (Fig. 3j), we investigated the possibility that MYC2 and GAME9 may form a TF complex that binds *GE1* to regulate *GAME18* expression through enhancer-promoter chromatin looping. Transactivation assays in tobacco protoplasts (Supplementary Fig. 6b) indicated that neither MYC2 or GAME9 alone, nor MYC2/GAME9 together are able to activate *LUC* expression driven by *proG18* (Fig. 6a). Remarkably, GAME9 was able to activate the *LUC* expression driven by *GE1-proG18* whereas MYC2 did not. Moreover, MYC2/GAME9 exhibited substantially higher activation ability compared to GAME9 alone (Fig. 6a). Taken together these data support the notion that the upregulation of *GAME18* by GAME9 is dependent on *GE1*, and that the MYC2/GAME9 TF complex potentiates the *GE1*-mediated transcriptional activity.

To narrow down the binding domains within the *GE1* enhancer, sub-fragments EI, EII and EIII were used to carry out transactivation assays (Supplementary Fig. 6b). None of MYC2, GAME9 or MYC2/GAME9 showed an ability to activate the *LUC* expression driven by EI or EII (Fig. 6b). Moreover, GAME9, but not MYC2, was able to significantly activate the expression of *LUC* reporter driven by EIII or GE1 (2.9-fold and 3.1-fold, respectively). Comparatively, MYC2/GAME9, showed the highest activation of *LUC* reporter driven by EIII or GE1 (3.9-fold and 4.3-fold, respectively) (Fig. 6b). These data suggest that EIII is the core *GE1* domain recruiting MYC2 and GAME9. The direct binding of both MYC2 and GAME9 to EIII via the G-box and GC-rich elements was further validated by both Y1H and EMSA assays

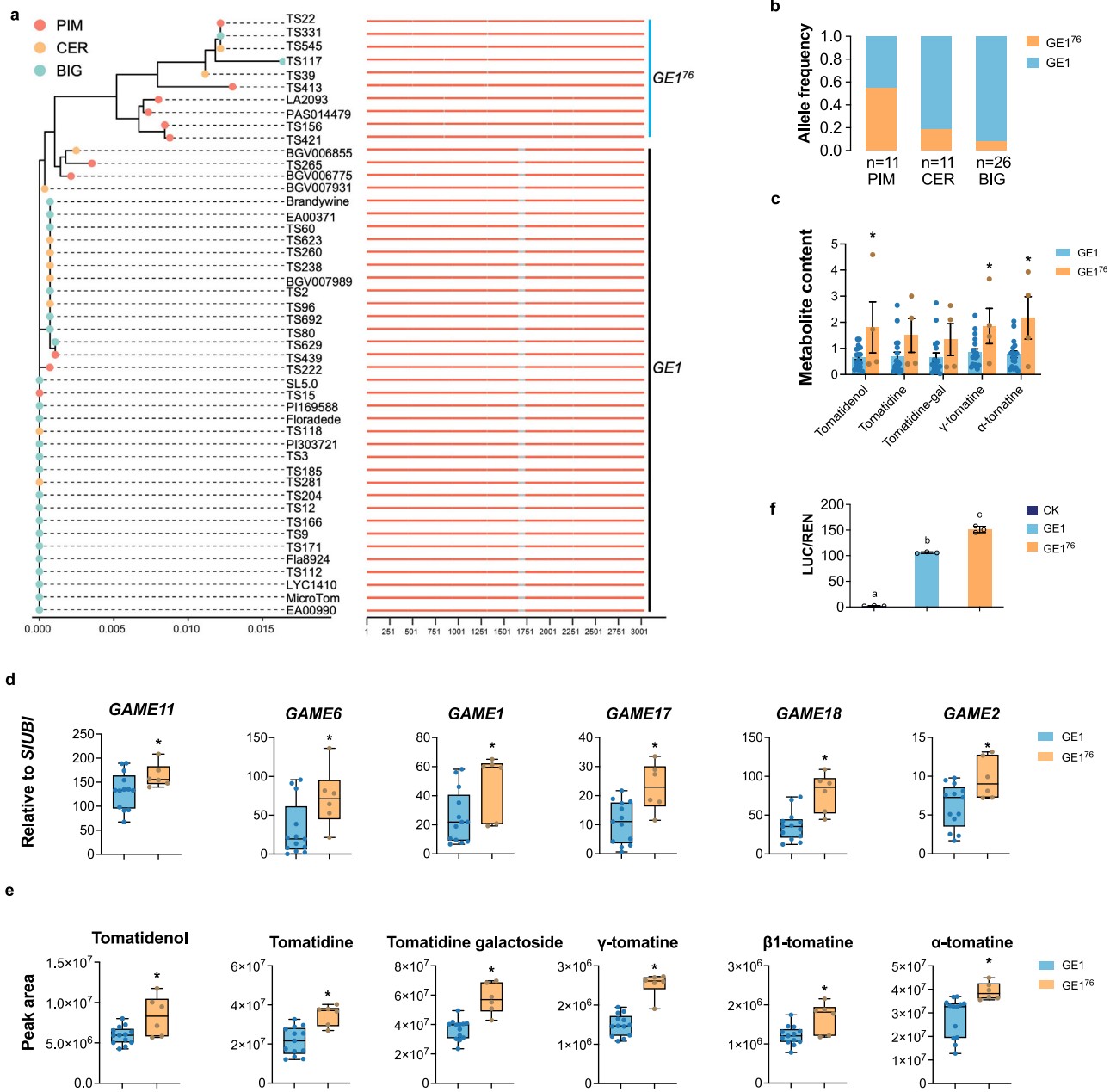

**Fig. 5 | Frequency of *GE1* and *GE1^{76}*, and the genetic effect. a** Phylogenic tree of 46 accessions, that was constructed with full-length of *GE1* by using RAxML software (100-times bootstrap) and GTRGAMMA model parameters. The gray lines represent the sequences that were deleted during domestication. **b** Allele frequencies of *GE1* in three groups. Data are presented as percentage of *GE1* in each groups. n, tomato accession numbers in group. PIM, *S. pimpinellifolium*. CER, *S. lycopersicum* var. *cerasiforme*. BIG, *S. lycopersicum*. **c** The five SGAs detected in 22 varieties were extracted from ref. 10. The metabolite content values of *GE1^{76}* were scaled to the *GE1* values. Dots indicate individual tomato accession, and data are presented as mean ± SEM ($n_{GE1} = 18$; $n_{GE1^{76}} = 4$). *$P < 0.05$ in two-sided $t$ test. **d** RT-qPCR analysis showing the expression levels of six *GAME* genes clustered on chr07 in 18 varieties.

Nineteen varieties with 13 contain *GE1* accessions (TS2, TS3, TS9, TS15, TS60, TS96, TS118, TS166, TS171, TS185, TS204, TS256, and TS439) and other 6 contain *GE1^{76}* accessions (TS22, TS39, TS117, TS156, TS413, and LA2093). Dots indicate individual tomato accession ($n_{GE1} = 13$; $n_{GE1^{76}} = 6$). *$P < 0.05$ in two-sided $t$ test. **e** SGAs profiles of nineteen tomato varieties of *GE1-* and *GE1^{76}*-containing accessions. Dots indicate individual tomato accession ($n_{GE1} = 13$; $n_{GE1^{76}} = 6$). *$P < 0.05$ in two-sided $t$ test. For all boxplots (**d**, **e**), box edges represent the 0.25 and 0.75 quantiles, the bold lines indicate median values and whiskers indicate 1.5× the interquartile range. **f** Transient assays showing *GE1^{76}* displayed a higher enhancer activity than *GE1*. Data are presented as mean ± SD ($n = 3$). *$P < 0.05$ in two-sided $t$ test.

(Fig. 6c–e). Interestingly, EMSA showed that the binding affinity of GAME9 to EIII was relatively weak, in contrast to the strong binding affinity of MYC2 to this *GE1* domain (Fig. 6d, e and Supplementary Fig. 6d). These data reveal the ability of both MYC2 and GAME9 to bind *GE1* via the EIII domain and that MYC2 enhances the binding affinity of MYC2-GAME9 to *GE1* which potentiates the transcriptional activity of the complex.

### MYC2 acts as a key mediator of GAME9-*GE1* interaction
Although MYC2 alone is unable to enhance *LUC* expression driven by *GE1* or *GE1-proG18*, its co-expression with GAME9 results in higher transcriptional activation than with GAME9 alone, indicating that MYC2 and GAME9 may form a TF complex that regulates *GAME18* expression through promoting the binding to *GE1*. In support of this idea, Y2H assays showed that MYC2 physically interacts with GAME9

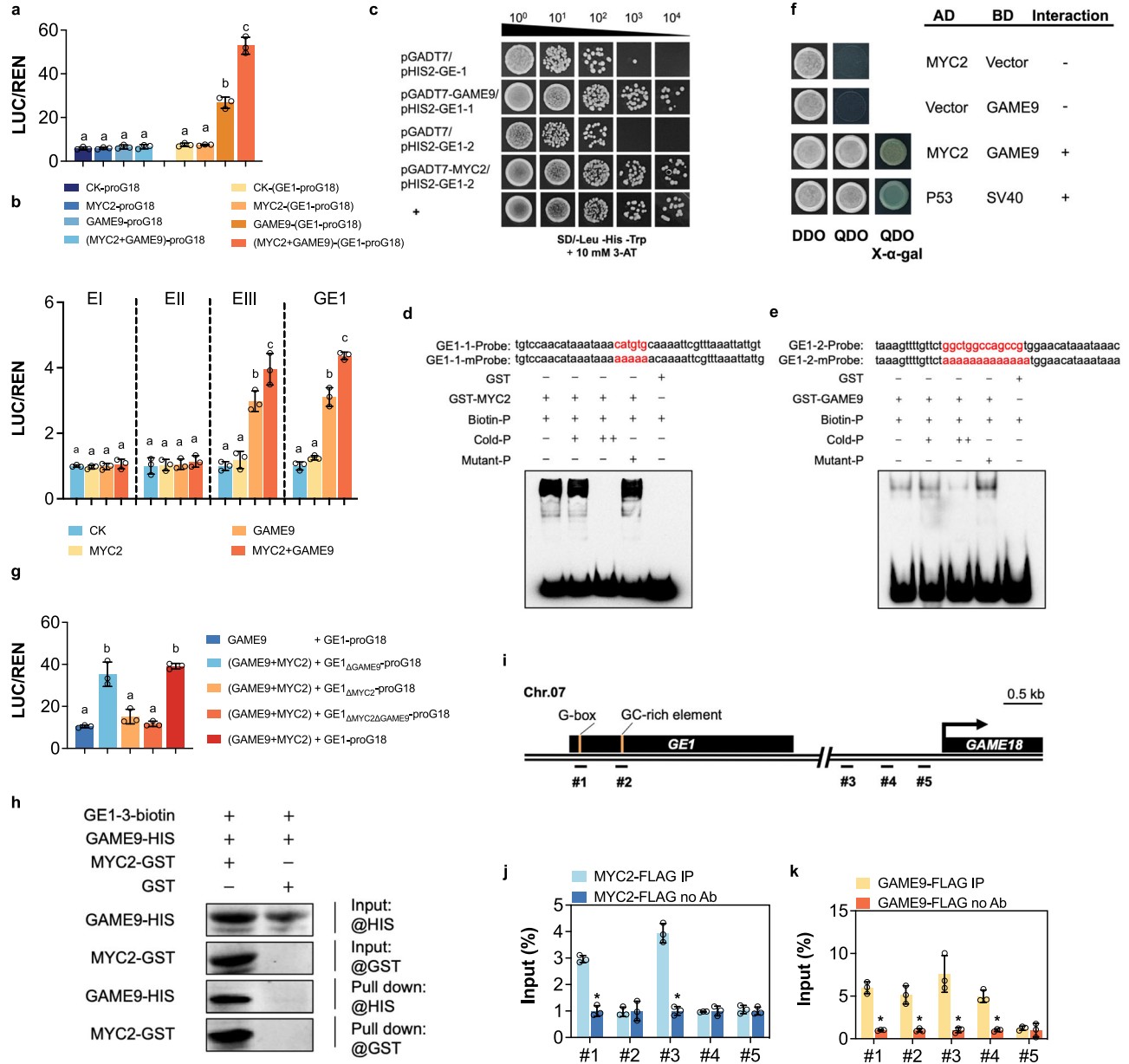

**Fig. 6 | Enhancer effects of *GE1*, and the binding of MYC2 and GAME9 to *GE1*.**
**a** Transient assays showing the transcriptional ability of MYC2 and GAME9 on *proG18* and *GE1-proG18* fusion fragment. **b** Transient expression assays showing the transcriptional ability of MYC2 and GAME9 on the GE1 and three fragments (EI, EII and EIII). **c** Yeast one-hybrid (Y1H) assays revealed the binding of MYC2 and GAME9 to *GE1* in vivo. EMSA assays showing the direct binding of MYC2 (**d**) and GAME9 (**e**) to the *GE1* via the G-box or GC-rich element. The sequence of GE1-1 and GE1-2 containing the G-box, and two GCC-rich element (highlighted in red), respectively. Biotin-P, the probe labeled with biotin. Cold-P, unlabeled probe. Mutant-P, mutated probe. The symbols - and + represent absence and presence, respectively, and + + indicates increasing amounts. The picture is representative of at least 3

independent assays all showing similar results. **f** Yeast two-hybrid (Y2H) assays between MYC2 and GAME9. P53 and SV40 vectors were co-transformed as positive controls. **g** Transient assays showing MYC2 was a key mediator for GAME9-*GE1* interaction. **h** DNA pull-down assay showing GAME9 bind to *GE1* through MYC2. GE1-3, the truncated GE1 fragment containing the MYC2 binding site was labeled with biotin. **i** Schematic diagram of *GE1* and *proG18* positions used in ChIP-qPCR assays. ChIP-qPCR of MYC2-FLAG (**j**) and GAME9-FLAG (**k**) levels at five sites (#1-#5) in MYC2-OE or GAME9-OE fruits at 7 DPA. Data are presented as mean ± SD ($n = 3$). *$P < 0.05$ in two-sided *t*-test. For all transient assays (**a, b, g**), data are presented as mean ± SD ($n = 3$). There are significant differences between the different letters (i.e., **a, b** etc) above the bars. $P < 0.05$ in one-way ANOVA.

(Fig. 6f). To further explore how the MYC2-GAME9 complex interacts with *GE1* to modulate the *GAME18* expression, we generated a series of *GE1-proG18* constructs mutated in the binding sites of either MYC2 (*GE1*$_{\Delta MYC2}$-*proG18*), or GAME9 (*GE1*$_{\Delta GAME9}$-*proG18*) or of both MYC2/GAME9 (*GE1*$_{\Delta MYC2\Delta GAME9}$-*proG18*) within the *GE1* fragment (Supplementary Fig. 6c). Remarkably, MYC2/GAME9 had a similar activating effect on *LUC* reporter expression driven by *GE1-proG18* and *GE1*$_{\Delta GAME9}$-*proG18*, in contrast to its failure to promote *LUC* expression driven by *GE1*$_{\Delta MYC2}$-*proG18* or by *GE1*$_{\Delta MYC2\Delta GAME9}$-*proG18* (Fig. 6g). The

requirement for a functional MYC2 binding motif clearly suggests that MYC2 is the key mediator within the MYC2-GAME9 complex, probably through increasing the affinity of the TF complex to the *GE1* enhancer.

To further validate this conclusion, we performed DNA pull-down assays with a biotin-labeled truncated *GE1* fragment (GE1-3) having a valid MYC2-binding site. The data indicates that the truncated fragment pulled down GAME9 only in the presence of MYC2 (Fig. 6h and Supplementary Fig. 6d), thus providing additional evidence that the binding of MYC2-GAME9 TF complex to *GE1* relies primarily on MYC2.

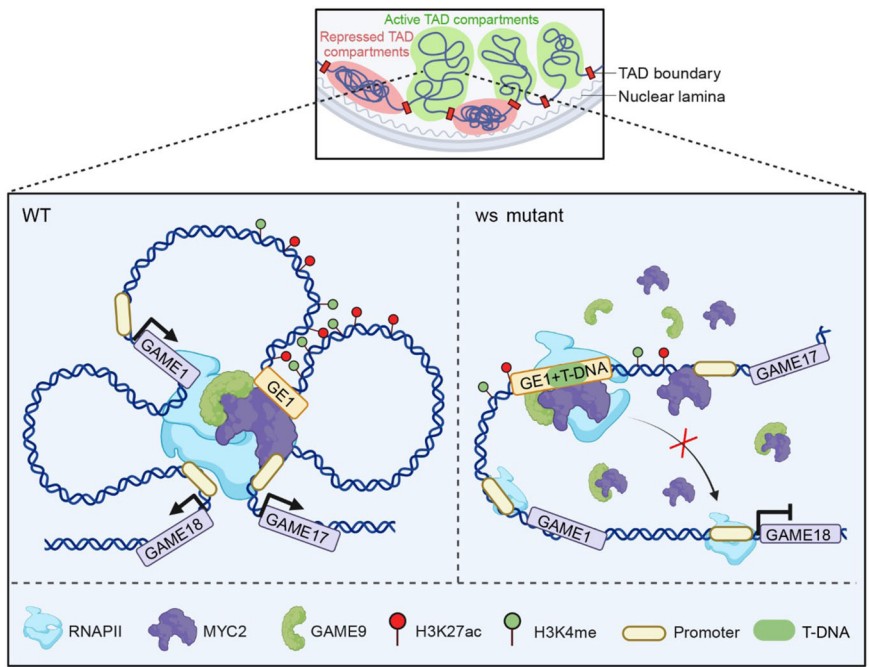

**Fig. 7 | The model of *GE1*-mediated regulation of expression of *GAME* cluster.** In the WT fruit (left), *GE1* can recruit MYC2 and GAME9 along with RNAPII to form a transcriptional complex that physically and functionally interacts with the *GAME* cluster though chromatin loop to dynamically regulate the expression of *GAME* cluster during fruit development. This process depends on the high levels of chromatin accessibility and histone modification (H3K27ac and H3K4me3). In the *ws* mutant fruit (right), T-DNA insertion disrupted the "TFs-enhancer-promoter" regulatory mechanism and led to the downregulation of *GAME18*, which caused the excessive accumulation of γ-tomatine, and eventually the wrinkled surface. TAD, topologically associating domain. The figure was created with BioRender.com.

Further evidence for the in vivo binding of MYC2 and GAME9 to *GE1* was provided by the ChIP-qPCR approach showing that more than 3-fold enrichments of MYC2 on region #1 and #3, and more than 4-fold enrichments of GAME9 on regions #1, #2, #3 and #4 were observed (Fig. 6i-k). In addition, mining the ChIP-seq data previously published for MYC2[26] identified a strong MYC2 signal on *GE1*, further validating the binding of MYC2 to *GE1* in vivo (Supplementary Fig. 6e). Altogether, the outcome of this study clearly demonstrates the role of *GE1* in regulating SGAs metabolism genes and uncovers its mode of action showing that the MYC2-GAME9 TF complex physically interacts with target genes through chromatin enhancer-promoter loops via direct binding to *GE1*.

## Discussion

The present study describes the role of *GE1*, an unprecedentedly reported distal enhancer, in regulating the GAME gene cluster underlying SGAs metabolism. The "TFs-enhancer-promoter" regulatory mechanism allows the accumulation of α-tomatine to defend against pathogens and herbivores at green stage (Fig. 7), which is gradually switched off during ripening to ensure that toxic α-tomatine can be converted into non-toxic and non-bitter esculeosides, making the fruit palatable for dispersal and reproduction. In brief, chromatin conformation, epigenetics and MYC2-GAME9 TF complex contribute to the shift in the SGAs profile. T-DNA insertion disrupted the "TFs-enhancer-promoter" regulatory module resulting in excessive accumulation of γ-tomatine and in a wrinkled surface phenotype of tomato fruits (Fig. 7). We show how a naturally occurring weak allele of this enhancer has been inadvertently selected during tomato domestication guided by the objective of reducing toxic SGAs levels in modern cultivars. Indeed, although high levels of the specialized metabolite γ-tomatine confer protection to plants from herbivory and pathogen invasion, the toxicity of this compound and its bitter taste make it unsuitable for human consumption. Our data indicate that the occurrence of *GE1*[76], the strong allele of the distal enhancer, dramatically decreases from the wild species through the intermediate *S.*

*lycopersicum* var. *cerasiforme* and to the modern *S. lycopersicum* big cultivars. Concomitantly, the content of the main SGAs is reduced in *GE1*-containing accessions bearing the weak allele compared to the *GE1*[76]-containing accessions.

Deciphering the operating mechanism of the *GE1* distal enhancer revealed that it recruits MYC2-GAME9 transcription factors complex to physically bind the promoter of *GAME* genes and to promote their expression through an enhancer-promoter chromatin loop structure engaging the *GAME* cluster genes. Combined ChIP-PCR, pull-down and transient expression approaches indicated that the binding of MYC2-GAME9 TF complex to *GE1* relies primarily on MYC2, which resolves the question of why none of MYC2 or GAME9 TFs is able to bind alone the promoters of *GAME* genes, although they have been clearly reported to regulate their expression. Interestingly, the sequences corresponding to the MYC2 and GAME9 binding motifs in the *GE1* region are conserved among the 46 accessions (Supplementary Data 8). This is in line with the studies in animals that enhancers are not always conserved, but their functions are during evolution[27].

Metabolic gene clusters involved in the biosynthesis of secondary metabolites are common in eukaryotes and prokaryotes[24,27,28]. We show that the *GAME* genes cluster located in tomato chr07 is confined within a TAD and wrapped in a chromatin loop involving the distal enhancer *GE1*. Our findings are in accordance with the recent study reporting that the thalianol gene cluster is located within a TAD and can be modulated by a super enhancer (SE) in *Arabidopsis*[21]. Unlike SE which acts as a super enhancer in regulating thalianol gene cluster, *GE1* alone can modulate the expression of GAME cluster, although we cannot rule out that the expression of *GAME* genes might be also regulated by other enhancers since *GAME* genes cluster were in multiple chromatin loops. Taken together, our study shows that besides the plant model Arabidopsis, enhancers serve as key regulators of the expression of metabolic gene clusters in crop species like tomato[16,21,29,30]. It is important to mention that in the particular case of tomato, the selection of advantageous allelic variants at the level of *GE1* enhancer sequence during tomato domestication was one of the

means of reducing the levels of anti-nutritional and toxic compounds in this widely consumed fruit nowadays.

Strikingly, while both the *ws* mutant carrying a T-DNA insertion and the *GE1* knockout lines exhibit similar phenotypes with high accumulation of toxic SGAs, the expression profile of GAME cluster genes is different. In *ws* mutant, *GAME1* and *GAME17* are upregulated and *GAME18* is downregulated, whereas in *GE1* knockout lines the three genes are downregulated. A possible explanation is that the loss of spatial interaction between *GE1* and GAME cluster in *ws* mutant may lead to increased levels of *GE1* eRNA which would preferentially target the genes closest to the enhancers, i.e., *GAME1* and *GAME17* thus resulting in their upregulation according to the "Enhancer release and retargeting (ERR)" model described previously[29]. By contrast, in *ge1* mutants, there is no production of functional eRNA due to the knockout of *GE1* which leads to the loss of regulation of *GAME* genes by the enhancer.

Overall, the identification and understanding of *cis*-regulatory modules (CRMs) is instrumental to the deciphering of the regulation mechanisms underlying complex developmental and metabolic processes[27]. CRMs are emerging as key components contributing to the high order complexity and extreme finesse of the regulatory processes that govern living organisms. In this regard, our findings provide important clues to understanding the functional significance of enhancer-promoter communication in regulating a complex metabolic pathway like SGAs cluster genes.

## Methods

### Plant materials

Tomato plants (*S. lycopersicum cv* MicroTom; *GE1* accessions including TS2, TS3, TS9, TS15, TS60, TS96, TS118, TS166, TS171, TS185, TS204, TS256, and TS439; *GE1⁷⁶* accessions including TS22, TS39, TS117, TS156, TS413, and LA2093) were grown in a climate-controlled greenhouse (Sichuan University, Chengdu, China) under the conditions of 14-h day/10-h night photoperiod, 23 °C during the day and 20 °C during the night, 50% relative humidity and 250 mol·m$^{-2}$·s$^{-1}$ intensity light.

Seeds of all *GE1* and *GE1⁷⁶* varieties used in this study were obtained from the Institute of Vegetable and Flower Research, Chinese Academy of Agricultural Sciences (IVF-CAAS). The *ws* mutant was obtained by an attempt to overexpress the CaMV 35S-driven MADS-box gene (*AGL66*, Solyc07g052700) in the MicroTom tomato cultivar.

### Hybridization and genotyping

For genetic analysis, WT (MicroTom) were crossed to ws lines to generate heterozygote lines (F$_1$ generation), and F$_1$ plants were selfed to obtain F$_2$ plants. Genotyping of T-DNA insertion was performed by analyzing of cloned PCR products in F$_1$ and F$_2$ plants. To isolate homozygous plants, DNA was extracted from F$_1$ and F$_2$ plants and genotyping was performed using three primers (LP, BP, and RP, listed in Supplementary Data 11) that amplified the T-DNA and flanking sequence. All F$_1$ plants showing two PCR products, indicating that were heterozygous. The F$_2$ generation plants were also genotyped by PCR, 1/4 F$_2$ plants showed a short DNA fragment (homozygous), 1/4 plants showed a long DNA fragment (WT), and 2/4 plants showed two fragments (heterozygous), showing the typical Mendelian segregation 1:2:1 for a single copy of T-DNA insertion in the transgenic tomato genome.

### Construction of plasmids and plant transformation

To construct plasmids *35S::GAME9*, and *35S::MYC2*, the coding sequences of *GAME9* and *MYC2* with a 3xFLAG were cloned into the pBI121 vector under the control of the CaMV 35 S promoter. The *GAME18* CRISPR/Cas9 constructs were cloned into the pFASTCas9/ccdB binary vector by GoldenBraid cloning system under the control of the *Arabidopsis* U6-26 consensus promoter. For the CRISPR/Cas9 construct carrying six gRNAs targeting the region of *GE1*, six potential 20 bp sites were selected for gRNA design within a region of 2922 bp

using a new multiplex CRISPR/Cas9 system[31]. The *GE1* CRISPR/Cas9 constructs were cloned into the pEx-6XsgR-Cas9 vector. Guide RNAs (gRNA) were designed using the CRISPR-P v.2.0 online tool (http://crispr.hzau.edu.cn/cgi-bin/ CRISPR2/CRISPR).

The final constructs pBI121-GAME9-FLAG, pBI121-MYC2-FLAG, pFASTCas9/ccdB-GAME18, and pEx-6XsgR-Cas9-GE1 were transformed into the tomato cultivar MicroTom via *Agrobacterium*-mediated transformation. Stable inheritance of transgenic plants was followed until the T$_2$ generation in overexpression lines and homozygous mutants for further analysis. All primers used in this study are listed in Supplementary Data 11 and all the sequences used are listed in Supplementary Data 12.

### Whole genome resequencing and analysis

Genomic DNA was extracted from 7-DPA tomato fruits using a Plant Genomic DNA Kit (TIANGEN; #DP305) and sequenced on Illumina HiSeq2000 platform at Novogene (Beijing, China). For obtaining clean, high-quality data, quality control of raw data from the Illumina HiSeq 2000 platform was checked using FastQC (www.bioinformatics.babraham.ac.uk/projects/fastqc). The reads were fed into fastp v0.22.0[32] to adapter trimming, low-quality reads removals and short-reads deletion. Next, the clean data was aligned to the tomato reference genome (SL4.0, https://solgenomics.net/ftp/genomes/Solanum_lycopersicum/assembly/build_4.00/S_lycopersicum_chromosomes.4.00.fa) and the pBI121-AGL66-FLAG vector sequence using BWA 0.7.17[33], respectively, to identify junction reads. Subsequently, the location information of the T-DNA insertion site was determined by the junction reads that aligned simultaneously with both the reference genome sequence and the vector sequence.

### RNA extraction and RT-qPCR

Total RNA from leaves and the pericarp of fruit at various stages during ripening was extracted using Plant RNA Purification Reagent (Invitrogen, Waltham, MA, USA; 12322-012). 1000 ng of total RNA used for first-strand cDNAs with Omniscript Reverse Transcription kit (Takara, Shiga, Japan; RR047). Gene-specific oligonucleotides were designed with Primer Express software (PE-Applied Biosystems, Waltham, MA, USA). RT-qPCR was performed as described in ref. 34., using the Bio-Rad CFX384 Real-Time PCR System (BIO-RAD, Hercules, CA, USA) with SYBR Green qPCR Mix (Vazyme, Nanjing, China; Q431-02). Three independent biological replicates were used for each sample. The data was analyzed by comparative threshold cycle method (ΔΔCt). *Actin* (Solyc11g005330) gene was used as internal control.

### RNA-seq analysis

Transcriptome profiling was performed as described in ref. 32. Briefly, total RNA was extract from 7-DPA fruits for the WT and *ws* lines with three independent biological replicates and the libraries were constructed and sequenced at Novogene (Beijing, China) with an Illumina NovaSeq 6000 instrument. For obtaining clean, high-quality data, quality control of raw data was checked using FastQC (www.bioinformatics.babraham.ac.uk/projects/fastqc). RNA-seq reads were fed into fastp v0.22.0[32] to adapter trimming, low-quality reads removals and short-reads deletion. Then, HISAT2 v2.2.1[35] was used to align the clean reads against the tomato reference genome with default parameter. Next, uniquely mapped reads with MAPQ > 15 was obtained by Samtools v1.13[36] to further analysis. The mapped reads for each gene were counted by featureCounts and transcripts per million (TPM) were calculated subsequently.

### Scanning electron microscopy (SEM)

Sections of 7 DPA fruit pericarp were fixed on a holder with a layer of carbon-rich conductive glue and critical point drying was performed using carbon dioxide[37]. The samples then transferred in the high vacuum cryo-unit, thin sputtered with a thin layer of gold–palladium,

and further inserted into the observation chamber with a rod. Microscopic observations of the fruit pericarp using a scanning electron microscope (JEOL; #JSM7500F; Sichuan university) with an acceleration voltage of 15 kV.

## Sample extraction and metabolomics analysis

Preparation of extracts and the profiling of SGAs in tomato tissues, including leaves and the pericarp of fruits were performed as described in refs. 3,7,38. Briefly, after freeze-dried, the samples were ground into fine power using automatic sample rapid grinder (50 Hz, 1 min, repeat 3 times). 0.1 g freeze-dried tissues were extract with 1 mL 80% methanol containing 0.1% formic acid (1:10 weight/volume radio), vigorously vortexed 30 s and sonicated for 30 min at 4 °C, followed by vortexing again for 30 s and 10 min of centrifugation at 20,000 g. The supernatant was filtered through a 0.22 μm polytetrafluoroethylene membrane filter and analyzed using the SCIEX Triple Quad™ 5500 LC-MS/MS System with the Ultra-High Performance Liquid Chromatography coupled with Mass Spectrometry (UPLC-MS) column connected online to a photo diode array detector (Shimadzu), separation of metabolites and detection of the eluted compound masses, mobile phases A and B, column temperature, solvent flow and injection volumes were as described. All samples were performed in biological triplicate. The relative quantification of SGAs metabolites was performed using α-tomatine as standard.

## High-through chromosome conformation capture (Hi-C) sequencing and analysis

Hi-C was performed on tomato fruits from WT and *ws* at 7 DPA as described in refs. 39,40. High-quality Hi-C sequencing libraries constructed by a 4-cutter restriction enzyme *Dpn*II were prepared. The Hi-C libraries were amplified by 12–14 cycles PCR, and sequenced in Illumina HiSeq platform. Sequencing interacting pattern was obtained by Illumina HiSeq instrument with $2 \times 150$-bp reads. After adaptor trimming, low quality and short reads removal using fastp v0.22.0, then read ends were paired and deduplicated. Juicer[41] was used to align clean reads against tomato reference genome (SL4.0) to construct contact matrices (MAPQ ≥ 30). Based on DNA fragment size, GC content and other genomic features, and comparing the observed interactions with the estimated expected interactions, we finally defined significant interactions as $P < 1 \times 10^{-15}$. Next, arrowhead with parameter '-k 10 kb' was used to obtain TADs and Hiccups[42] with parameters '-m 512 -r 5000,10000 -k KR -f .1,.1 -p 4,2 -i 7,5 -t 0.02,1.5,1.75,2 -d 20000,20000' were used to chromatin loops. HiC heatmap and chromatin loops map were visualized by HiCExplore[43].

## Transient assay

To test the enhancer activity, *GE1*, *GE1_{T-DNA}* (495 bp T-DNA sequence was inserted at the T-DNA insertion site to mimic *GE1* in *ws*), *GE1*[76] (the variant of *GE1*) and truncated segments of *GE1* (EI, EII, and EIII) were separately cloned into *PGL3* vector with Minimal promoter *35S* (*mp35S*)-driven Luciferase (*LUC*). To further exam the enhancer activity to target genes, the *GAME18* promoter (*proG18*), the *GAME1* promoter (*proG1*), and the *GAME17* promoter (*proG17*) linked to *mp35S* following these fragments (EI, EII, EIII and GE1) were constructed as reporter plasmids.

For the transactivation assays to test the regulation of MYC2 and GAME9 on the enhancers, full-length CDS of GAME9 and MYC2 were individually cloned into the pGreenII 62sk vector with an CaMV 35S promoter were constructed as effectors. To further explore the functions of MYC2 and GAME9 proteins in the transcription complex, the variant *GE1_{ΔMYC2}* (mutated the MYC2 bonding site), *GE1_{ΔGAME9}* (mutated the GAME9 bonding site), and *GE1_{ΔMYC2ΔGAME9}* (mutated both the MYC2 and GAME9 bonding sites) were cloned into *PGL3* vector as reporters. *pRL-null* vector was used as the internal control, which had the *mp35S*-driven Renilla Luciferase (*REN*).

Seedings of tobacco were grown in the culture room at 28 °C for 20–30 days, and the leaves were harvested to isolate mesophyll protoplasts. Transfection was performed as described in ref. 44, and dual-luciferase detection was conducted according to the manual of the Dual-Luciferase Reporter Assay System E1960 (Promega, Madison, WI, USA; E1910) at -14–16 h after transfection. For each assay, three independent biological replicates and two technological replicates were performed.

## Electrophoretic Mobility Shift Assays (EMSA)

The full coding sequence of GAME9 and MYC2 were amplified from MicroTom, and then inserted into *pGEX-4T-1* vector to produce GAME9-GST and MYC2-GST fusion proteins. The recombinant fusion protein with a GST-tag was expressed in Trans BL21 cells and *Rosetta* (DE3) *E. coli* cells (Transgene, Beijing, China) with 0.1 mM isopropyl-1-thio-D-galactopyranoside (IPTG) at 18 °C for 14 h, affinity purified using Glutathione Sepharose 4B (GE Healthcare, Chicago, IL, USA) according to manufacturer's instructions, that purified proteins were detected by 10% SDS-PAGE. The probes containing a G-box or GC-rich element derived from the *GE1* were synthesized with a 3'-biotin label and annealed to form double-stranded oligonucleotides. The unlabeled DNA fragment was used as competitor, and the mutated probes had the variant G-box or GC-rich element that had been changed to AAAAAA or AAAAAAA. The purified proteins (20–40 ng fusion protein per reaction) were incubated with probes in a binding buffer (10 mM Tris-HCl [pH 7.2], 50 mM KCl, 1 mM DTT, 2.5% glycerol, 0.05% NP-40, and 50 ng· μL⁻¹ polydeoxy [inosinate-cytidylate]) at room temperature for 20 min. Protein-DNA complexes were separated on 5% (w/v) native polyacrylamide gel electrophoresis and transferred to nylon membranes, and the biotin-labeled probes were detected according to the instructions provided with the EMSA kit (Thermo Fisher; 20148) for chemiluminescent detection. The sequence of the biotin-labeled probes was listed in Supplementary Data 11.

## DNA pull-down assay

To produce GAME9-HIS fusion protein, full-length CDS of GAME9 was PCR amplified and cloned into pET28a vector. The recombinant GST, MYC2-GST (from vector pGEX-4T-1), HIS and GAME9-HIS proteins were produced in BL21 codon plus Escherichia coli cells and purified. The truncated fragment of GE1-3 (500 bp), was amplified by PCR using 5-biotin-labeled primers. For DNA pull-down assay, GAME9-HIS protein (40 μg) was incubated with biotin-labeled DNA together with resin-bound GST or MYC2-GST fusion proteins in HKMG buffer (pH 7.9, 10 mM HEPES, 100 mM KCl, 5 mM MgCl2, 10% [v/v] glycerol, 1 mM DTT, and 0.5% [v/v] NP-40) containing protease and phosphatase inhibitors at 4 °C overnight. Streptavidin agarose beads (Sigma; 16-162) was used for pulled down the DNA-binding proteins, and samples detected by immunoblot analysis using an anti-GST antibody (1/1000, Cell Signaling Technologies, #2622) and anti-HIS antibody (1/3000, Cell Signaling Technologies, #9991).

## Chromatin immunoprecipitation (ChIP) followed by qPCR (ChIP-qPCR)

ChIP assays were performed on WT, *ws*, *GAME9*-OE, and *MYC2*-OE lines as described in ref. 45. In brief, the fruits at different stages were harvested and cross-linked with 1% (v/v) formaldehyde under vacuum about 15 min, which was quenched in 0.125 M glycine under vacuum for 5 min. After three times washed, the quenched samples were quickly frozen in liquid nitrogen. Nuclei were isolated by resuspending the pulverized tissue in ice-cold Honda buffer (0.44 M sucrose, 20 mM HEPES, 5 mM DTT, 1 mM PMSF, 1.25% Ficoll 400, 2.5% Dextran 40, 0.5% Triton X-100, and protease and phosphatase inhibitor) and passage through two layers of miracloth (Millipore) at 4 °C. The chromatin was pelleted and resuspended in nuclei dilution buffer (1 M Tris-HCl [pH 8.0], 0.5 M EDTA, 10% SDS, 100 mM PMSF, and protease and

phosphatase inhibitors), and sonicated on Bioruptor system for 11 min (30% power, 10 s "ON" and 10 s "OFF") to shear DNA fragments of 200-800 bp long.

The chromatins were immunoprecipitated with protein A agarose beads and antibodies against FLAG (1/50, Cell Signaling Technologies, #14793), or H3K27ac (1/100, Cell Signaling Technologies; #8173), or anti-RNA polymerase II (1/500, Abcam, #ab 264350) and then incubated at 4 °C overnight. The beads were washed in low salt buffer (150 mM NaCl, 0.1% [w/v] SDS, 1% [v/v] Triton X-100, 2 mM EDTA, 20 mM Tris-HCl [pH 8.0]), then washed in high salt buffer (500 mM NaCl, 0.1% [w/v] SDS, 1% [v/v] Triton X-100, 2 mM EDTA, 20-mM Tris-HCl [pH 8.0]), and then washed in LiCl buffer (250 mM LiCl, 1% [w/v] sodium deoxycholate, 1% [v/v] NP-40, 1 mM EDTA, 10 mM Tris-HCl [pH 8.0]), and finally washed twice in TE buffer (1 mM EDTA, 10 mM Tris–HCl [pH 8.0]). Samples were incubated at 65 °C overnight for reversing crosslink, and then incubating with 50 mg/mL Proteinase K (Cell Signaling Technologies; #10012 S) at 65 °C for 3 h. The immuno-precipitated chromatin was recovered, dissolved in water for subsequent qPCR analysis.

### Phylogenetic tree
Phylogenetic tree of *GE1* was built after alignment of 10 kb sequence around *GE1* (5 kb upstream and 5 kb downstream) using MUSCLE algorithm and Maximum Likelihood clustering. The sequence data of *GE1* from 46 tomato accessions according to Ref. 46. Analyses were conducted using RAxML software applied with 100-times bootstrap[47].

### Statistical analysis
All experiments were repeated at least three times independently, and results from representative datasets are presented. Statistical analysis using unpaired two-tailed Student's *t*- test and one-way ANOVA. Related information is listed in the Source Data.

### Reporting summary
Further information on research design is available in the Nature Portfolio Reporting Summary linked to this article.

## Data availability
The whole genome resequencing data, the RNA-Seq data and the Hi-C sequencing data have been deposited in the Genome Sequence Archive (GSA) under accession number CRA011801. The DNase-seq data, and ChIP-seq data (H3K27ac, H3K4me3, H3K27me3 and MYC2) were downloaded from NCBI (BioProjects: PRJNA381300, SRA046131 and PRJNA375842)[23,24,26]. The source data for Figs. 1–6, and Supplementary Figs. 1–4 are provided as a Source Data file. Source data are provided with this paper.

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

## Acknowledgements

This work was supported in part by the National Natural Science Foundation of China (No. 31772372 and No. 32172643 to M.L. and No. 32172271 to H.D.), the Natural Science Foundation of Sichuan Province, China (2023NSFSC1991 to Y.Z.), the Institutional Research Funding of Sichuan University (2022SCUNL105 to M.L.), the Applied Basic Research Category of Science and Technology Program of Sichuan Province (2021YFQ0071; 2022YFSY0059-1; 2021YFYZ0010-5-LH to M.L.), and the Technology Innovation and Application Development Program of Chongqing (cstc2021jscx-cylhX0001 to M.L.). We thank Dr. Hsihua Wang from the Mass Spectrometry Core Facility in College of Life Sciences, Sichuan University for the assistance in metabolic analysis.

## Author contributions

M.L. planned and designed the research; F.B., P.S., H.D., Y.W., Y.C. and M.W. performed experiments. T.M., Y.Z., J.P. and Z.L. analyzed data. F.B., M.L. and M.B. wrote the manuscript and Y.H. helped improve the manuscript.

## Competing interests

The authors declare no competing interests.
