## [Peer Review File · Nature Communications]

Reviewers' comments:

Reviewer #1 (Remarks to the Author):

This manuscript describes the identification of an enhancer of GAME genes. This enhancer regulates the transcription of GAME genes by recruiting the MYC2-GAME9 complex. The results are interesting and worthy of publication. However, there are some problems with the presentation of the data and the interpretation of results. Specific criticisms are below.

Figure 3. The GE1 and GE1-76 terminology is confusing. Since *S. pimpinellifolium* is the ancestral genotype, shouldn't it be termed a deletion of 76bp in BIG varieties not an insertion in PIM varieties.

Figure 3. It would be nice to have the tomatine data for all varieties shown in the tree, not just the 23 with publicly available data.

Figure 3 d and e. Again, it would be nice to see data for all varieties shown in Figure 3a, rather than just four. How do you know that these 4 varieties are representative of the GE1 and GE1-76 genotypes?

Why does the ws mutant only show more β -tomatine, but the GE1 and CRISPR lines show more of all SGAs?

What exactly are EI, EII and EIII? How do they overlap with GE1? This should be explained in the Figure 2 a legend. This should also be pointed out in the text when E1, EII and EIII are discussed. The same should be done for P1, P2, P3, P4 and P5. What is CK in Figure 2d?

Figure 2 e-k: I don't see a red box and am confused which black lines are mentioned in the legend. Maybe the figure is just too small to see these features. I cannot see any differences between WS and ws. Again, Maybe the figure is too small to see the differences.

Are there error bars for figure 3C? Are the differences significant? Why are relative values shown rather than the actual values?

Does PIM mean *S. pimpinellifolium* in Figure 3 D and e? Why were only *S. pimpinellifolium* examined? It would be nice to show data for other varieties.

The legends for Figure 4 d, e and h are very confusing. How do the tables above relate to the figures? Why aren't the table labels the same as the figures in 4h?

Figure 4 d and e need more explanation. What are the DNA sequences above the EMSA? Which sequence was used for the binding. What are the various controls?

Line 262: MYC2-GMAE9 is this MYC2-GAME9

GAME is annotated as GMAE in Figure 6 and other places in the manuscript. Which is correct?

What does TAD stand for in Figure 6? It should be defined at the first mention of it in the manuscript.

I'm not sure what this sentence means; "Interestingly, the motif of MYC2 and GAME9 retention rate were stable across the 46 accessions compared with GE1 sequence identity."

This sentence needs a reference: " This is in line with the studies in animals that enhancers are not always conserved, but their functions are during evolution."

Figure 1f. The green and brown arrows are confusing. Does the direction of the area indicate up or down regulation in WS or ws fruit?

Figure 2a. DHS and H3K27ac need to be defined.

Extended figure 1a. What are AGL66-OE38 and AGL66-RNAi5?

Extended figure 1d and e. Does the column color have any significance?

Extended figure 3i. Why does the ws mutant have larger lesions than the WT, if the ws

Reviewer #2 (Remarks to the Author):

The authors report an "accidental" discovery of a distal enhancer, GE1, which regulates the expression of the tomato steroidal glycoalkaloid (SGA) genes. Insertion of a T-DNA near GE1 resulted in transcriptional alteration of multiple genes in the SGA gene cluster, which ultimately caused γ -tomatine accumulation and a "wrinkled fruit surface" phenotype. Remarkably, the T-DNA insertion caused down-regulation of GAME18, but up-regulation of GAME1 and GAME17. The authors conducted high-throughput chromosome conformation capture (Hi-C) analysis and identified a TAD spanning the SGA gene cluster. The putative interaction between GE1 and GAME18 is lost in the T-DNA line; however, the interactions between GE1 and GAME1/17 are retained (Figure 2). Additionally, the authors discovered a naturally occurring GE176 allelic variant, which is more active in stimulating the expression of GAME genes. A weaker GE1 allele is proposed to be the main driver for selecting reduced SGA levels during tomato domestication. It is quite interesting to see the role of GE1 in regulation of multiple GAME genes. However, I have major concerns for this manuscript:

1. It lacks genetic evidence of GE1 function. Deletion mutations, which can be readily developed using CRISPR/Cas in tomato, are absolutely required to provide direct evidence for the function of GE1 as well as the functional difference between GE1 and GE176. Most of the data in this manuscript, including DHS, histone modifications, and Hi-C, provide only correlative or indirect evidence for enhancer function.

2. The authors have ignored the extensive recent literature on the cis-regulatory function of DNA sequences associated with accessible chromatin regions (DHSs). Based on the literature, the authors should focus on the DHSs within GE1, rather than dividing GE1 into three different segments based on information of H3K27ac, which has recently been disapproved as an enhancer mark in *Arabidopsis thaliana* (Yan et al. 2019, Nature Communications).

3. The Hi-C data is not well described and illustrated with details. I am not able to see the interaction between the GE1 region and the individual GAME genes. In addition, the authors didn't explain why the T-DNA insertion only disrupts the contact on one side of GE1, and why the disruption from T-DNA would up-regulate (not down-regulate) GAME1 and GAME17?

4. The authors briefly cited a recent publication (Zhao et al. 2022, PNAS) of an enhancer in *Arabidopsis thaliana*, which similarly regulates multiple genes in a biosynthetic gene cluster, and used a very similar diagram in Figure 6 as Zhao et al. (Figure 5). However, there is no discussion about the similarity and difference of these two enhancers.

Point-by-point responses to Reviewer #1 (Remarks to the Author):

This manuscript describes the identification of an enhancer of GAME genes. This enhancer regulates the transcription of GAME genes by recruiting the MYC2-GAME9 complex. The results are interesting and worthy of publication. However, there are some problems with the presentation of the data and the interpretation of results. Specific criticisms are below.

Response: We thank the reviewer for the valuable and instructive comments to improve our manuscript.

Q: Figure 3. The GE1 and GE1⁷⁶ terminology is confusing. Since *S. pimpinellifolium* is the ancestral genotype, shouldn't it be termed a deletion of 76 bp in BIG varieties not an insertion in PIM varieties.

Response: The reviewer is right. It is more appropriate to refer to a deletion in BIG varieties than an insertion in PIM. Following the reviewer's suggestion, we have now termed *GE1*⁷⁶ in PIM varieties as the original version and *GE1* as the deletion version in our revised manuscript.

Q: Figure 3. It would be nice to have the tomatine data for all varieties shown in the tree, not just the 23 with publicly available data.

Response: We thank the reviewer for the valuable suggestion. Unfortunately, publicly available data are only available for 23 varieties but not for the other varieties presented in the phylogenetic tree which explains why we didn't show the tomatine data for those varieties. However, we believe that the data provided for 23 varieties which include both *GE1* and *GE1*⁷⁶-containing accessions, clearly show that the content of five main SGAs is higher in *GE1*⁷⁶-containing accessions than in *GE1*-containing accessions, supporting the hypothesis that *GE1* was an important driver in the process for reducing SGAs content during tomato domestication and that the weaker allele *GE1*, rather than *GE1*⁷⁶, has been selected in modern tomato cultivars.

Q: Figure 3 d and e. Again, it would be nice to see data for all varieties shown in Figure 3a, rather than just four. How do you know that these 4 varieties are representative of the *GE1* and *GE1*⁷⁶ genotypes?

Response: We agree with the reviewer that it would be interesting to see the expression levels of *GAME* genes and SGAs contents in all varieties. As mentioned above, we don't have a full set of data for all varieties. However, we obtained four PIM varieties among which two correspond to the *GE1*⁷⁶ variant (TS22 and TS156) and the other two contain the *GE1* allele (TS15 and TS439). While these four varieties were randomly selected, both the *GAME* expression levels and SGAs contents are relatively higher in the two *GE1*⁷⁶ accessions (TS22 and TS156) than in *GE1* accessions (TS15

and TS439) which supports the hypothesis that the weaker allele *GEI*, rather than *GEI*⁷⁶, has been selected in modern tomato cultivars.

Q: Why does the *ws* mutant only show more γ -tomatine, but the *GEI* and CRISPR lines show more of all SGAs?

Response: Not sure we get here the reviewer point. The *GAME18* CRISPR lines (*game18*) only exhibit high γ -tomatine content (Figure 1i in our revised manuscript), but the *ws* mutant showed more γ -tomatine and also increased levels of SGAs compounds corresponding to the downstream part of the pathway (Figure 1e in our revised manuscript). We guess the question is about the difference in SGAs profile between *ws* mutants and *game18* lines. In *ws* mutant, the *GAME* cluster genes were all affected due to the disruption of *GEI*-mediated chromatin loops resulting from the T-DNA insertion, while in *game18* lines, only *GAME18* was knocked out. So, the different expression profiles of *GAME* genes in *ws* mutant and *game18* lines might account for the differences in SGA contents.

Q: What exactly are EI, EII and EIII? How do they overlap with *GEI*? This should be explained in the Figure 2a legend. This should also be pointed out in the text when EI, EII and EIII are discussed. The same should be done for P1, P2, P3, P4 and P5. What is CK in Figure 2d?

Response: We have now explained more clearly this in the figure legends and in the main text of our revised version. As shown in Figure 2a, EI, EII and EIII are sub-fragments of the 2.9-kb *GEI*. EI, EII and EIII are located within the *GEI* region (2922 bp) at the following coordinates: 2032-2922 (EI), 943-1871 (EII), and 1-769 (EIII), respectively. To test which part of *GEI* (2.9 kb) function as the key fragment of enhancer, we segmented the 2.9-kb region into three sub-fragments (EI, EII and EIII) based on the presence of typical enhancer features (Figure 2a in our revised manuscript). The data showed that the *GEI* full fragment promotes higher transcriptional activity than EI, EII, or EIII sub-fragments, suggesting the need for all three sub-domains to have a fully functional *GEI*. On the other hand, P1, P2, P3, P4 and P5 refer to the five sites in *GEI* designed for ChIP-qPCR to examine RNAPII and H3K27ac levels at different regions of *GEI* in WT and *ws* mutant. The control (CK) (Figure 2f in our revised manuscript) refers to the empty pGL3 vector. We thank the reviewer for pointing out these issues and we have now revised the figure legends and the main text referring to these points which indeed improved the clarity of this part of our manuscript.

Q: Figure 2 e-k: I don't see a red box and am confused which black lines are mentioned in the legend.

Maybe the figure is just too small to see these features. I cannot see any differences between WS and ws. Again, Maybe the figure is too small to see the differences.

Response: We thank for the reviewer for pointing out this issue. The solid black line represents the topologically associating domains (TADs), and the red box represents the chromatin interactions regions. We have now provided new figures (Figure 3a-f in our revised manuscript) with improved resolution to better emphasize the features including the red box, black lines, and the differences in chromatin loops between WT and *ws* mutant (See figure below). We have also marked the solid black lines and red boxes using white and green arrows, respectively. The difference of chromatin loops between WT and *ws* mutant was marked with green circles.

Chromatin interaction heatmaps of chromosome 7 in WT and *ws* fruits at 100 kb (a and d), 10 kb (b and e), or 5 kb (c and f) resolution. The solid black line represents the topologically associating domains (TADs), and the red box represents the chromatin interactions region. Solid black lines and red boxes are marked by white and green arrows, respectively. The difference of chromatin loops between WT and *ws* mutant is marked with green circles.

Q: Are there error bars for figure 3C? Are the differences significant? Why are relative values shown rather than the actual values?

Response: Since the “actual values” from the publicly available data (Zhu *et al.*, 2018, *Cell*) are represented as peak area and the variations of the SGAs content in the 23 accessions including both *GE1* and *GE1*⁷⁶ accessions were very large, we used the relative values instead of the actual values (peak area) following the method used previously (Zhu *et al.*, 2018, *Cell*). We have now provided

a new figure with error bars (Figure 5c in our revised manuscript) without statistical analysis since the publicly available data for each SGAs relative content were just two biological replicates.

Q: Does PIM mean *S. pimpinellifolium* in Figure 3 D and e? Why were only *S. pimpinellifolium* examined? It would be nice to show data for other varieties.

Response: The reviewer is right. PIM refers to *S. pimpinellifolium* in Figure 5d and e in our revised manuscript (former Figure 3 d and e). The reason for the selection of the *S. pimpinellifolium* varieties for this experiment is that we only got seeds for four varieties all of which are PIM, we then tested the 4 PIM varieties which include 2 carrying the *GE1* allele and 2 the *GE1*⁷⁶ allele. We believe the reviewer's suggestion is valuable, and we hope we can be able to examine more varieties in the future.

Q: The legends for Figure 4 d, e and h are very confusing. How do the tables above relate to the figures? Why aren't the table labels the same as the figures in 4h? Figure 4 d and e need more explanation. What are the DNA sequences above the EMSA? Which sequence was used for the binding. What are the various controls?

Response: We greatly appreciate the reviewer's comments and suggestions regarding these issues. We have now modified the figure legends to add the details needed to better explain the figures (please note that the former Figure 4 is now Figure 6 in our revised manuscript). We include the following sentence "The sequence of GE1-1-Probe contains the G-box (MYC2 binding site, highlighted in red). GE1-1-mProbe is the mutated probe in which the G-box motif 5'-catgtg-3' was replaced by 5'-aaaaaa-3' (highlighted in red). The sequence of GE1-2-Probe contains two consecutive GCC-rich element (GAME9 binding site, highlighted in red). GE1-2-mProbe is the mutated probe in which the GCC-rich element motif 5'-ggctggccagccg-3' was replaced by 5'-aaaaaaaaaaaa-3' (highlighted in red). Biotin-P, the probe labeled with biotin. Cold-P, unlabeled probe. Mutant-P, mutated probes. The GST protein was incubated with the labeled probe to serve as negative control. 10× and 100× Cold-P, and Mutant-P were used for competition".

In Figure 6h, the length of GE1-3 (for DNA Pull-down) used is about 500 bp, compared to the 50 bp of the GE1-1 and GE1-2 fragments (for EMSA) used in figure 6d and 6e. For this reason we did not show the table labels in Figure 6h as we did in figure 6 d and e. We have now provided a new extended data figure (Extended Data Fig. 6d) to clearly show the fragments in *GE1* used in Figure 6d (GE1-1), 6e (GE1-2), and 6h (GE1-3). Please note that the original Figure 4 is now Figure 6 in our revised manuscript.

The sequence of the DNA fragment used for EMSA in Figure 6d correspond to a portion of *GE1*

(about 50 bp) containing the G-box (MYC2 binding site, highlighted in red), while the sequence of the DNA fragment used for EMSA in Figure 6e is another portion of *GEI* (about 50 bp) harboring the GCC-rich element (GAME9 binding site, highlighted in red). The sequence of the DNA fragment used for DNA Pull-down in Figure 6h is a portion of *GEI* (about 500 bp) containing the G-box (MYC2 binding site). The details of the controls used in these figures have been now added in the figure legends.

Q: Line 262: MYC2-GMAE9 is this MYC2-GAME9. GAME is annotated as GMAE in Figure 6 and other places in the manuscript. Which is correct? What does TAD stand for in Figure 6? It should be defined at the first mention of it in the manuscript.

Response: We apologize for these typo mistakes and thank the reviewer for pointing out these issues. GMAE is a typo mistake for GAME. We have corrected these typos throughout the manuscript. In Figure 7 (former Figure 6), TAD stands for Topologically Associating Domain, we have now defined it at the first mention in the manuscript and in the figure legend.

Q: I'm not sure what this sentence means; "Interestingly, the motif of MYC2 and GAME9 retention rate were stable across the 46 accessions compared with GE1 sequence identity."

Response: The sentence have been changed as follows: "Interestingly, the sequences corresponding to the MYC2 and GAME9 binding motifs in *GEI* region are conserved among the 46 accessions". Based on the data shown in Supplementary Dataset 8, we found that although the sequences of *GEI* displayed a change (such as from *GEI*⁷⁶ to *GEI*) during domestication, the sequences of the binding sites for both MYC2 and GAME9 are relatively conserved. This suggested that although the enhancer activity of *GEI* may change, their regulation of the GAME gene cluster depends on the recruitment of MYC2-GAME9 transcriptional complex, supporting the idea that the "TF-enhancer-promoter" regulatory mode is conserved.

Q: This sentence needs a reference: "This is in line with the studies in animals that enhancers are not always conserved, but their functions are during evolution."

Response: We have now cited the following reference for this sentence. Thanks.

Wong, E. S. *et al.* (2020). Deep conservation of the enhancer regulatory code in animals. *Science* **370**, eaax8137.

Q: Figure 1f. The green and brown arrows are confusing. Does the direction of the area indicate up or down regulation in WT or ws fruit?

Response: The reviewer is right. The direction of the arrows indicates up or down regulation of gene expression in *ws* compared to WT. The different colors of the arrows are meant to better distinguish between up (brown) or down (green) regulation. We have now clarified this in the figure legends.

Q: Figure 2a. DHS and H3K27ac need to be defined.

Response: We have defined DHS (DNase I hypersensitivity site) and H3K27ac (Histone H3 acetylated lysine 27) in the figure legends.

Q: Extended figure 1a. What are AGL66-OE38 and AGL66-RNAi5?

Response: We have now made it clear in the figure legend that *AGL66*-OE38 corresponds to *AGL66* overexpression (OE) line and *AGL66*-RNAi5 to *AGL66* downregulated line expressing an RNAi construct. When overexpressing the MADS-box gene *AGL66* (Solyc07g052700) in tomato, we obtained a single overexpressing (*AGL66*-OE8) line with wrinkled surface fruit which is different from all other OE lines obtained. Assessment of the relative expression level of *AGL66* by RT-qPCR excluded the expression dose effect since the transcript levels of *AGL66* in *AGL66*-OE8 is not the highest among all *AGL66*-OE lines. Moreover, downregulation of *AGL66* by RNAi resulted in no difference regarding the fruit surface compared to WT. *AGL66*-OE38 line shows the highest up-regulation of *AGL66* among all OE lines, and *AGL66*-RNAi5 shows the strongest down-regulation among all RNAi lines. These results suggested that the fruit wrinkled surface phenotype of *AGL66*-OE8 (wrinkled surface mutant, *ws*) does not correlate with the altered expression of *AGL66* but is likely due to the T-DNA insertion event which impairs a functional locus that is not related to *AGL66*.

Q: Extended figure 1d and e. Does the column color have any significance?

Response: In figure 1d and e, the column colors stand for different transgene lines. To avoid any confusion, we modified all the columns to the same color.

Q: Extended figure 3i. Why does the *ws* mutant have larger lesions than the WT, if the *ws*

Response: Apparently the sentence is truncated which doesn't allow us to get the full question of the reviewer.

References

Zhu, G. *et al.* (2018). Rewiring of the Fruit Metabolome in Tomato Breeding. *Cell* **172**, 249-261.

Wong, E. S. *et al.* (2020). Deep conservation of the enhancer regulatory code in animals. *Science* **370**, eaax8137.

Point-by-point responses to Reviewer #2 (Remarks to the Author):

The authors report an “accidental” discovery of a distal enhancer, *GEI*, which regulates the expression of the tomato steroidal glycoalkaloid (SGA) genes. Insertion of a T-DNA near *GEI* resulted in transcriptional alteration of multiple genes in the SGA gene cluster, which ultimately caused γ -tomatine accumulation and a “wrinkled fruit surface” phenotype. Remarkably, the T-DNA insertion caused down-regulation of *GAME18*, but up-regulation of *GAME1* and *GAME17*. The authors conducted high-throughput chromosome conformation capture (Hi-C) analysis and identified a TAD spanning the SGA gene cluster. The putative interaction between *GEI* and *GAME18* is lost in the T-DNA line; however, the interactions between *GEI* and *GAME1/17* are retained (Figure 2). Additionally, the authors discovered a naturally occurring *GEI*⁷⁶ allelic variant, which is more active in stimulating the expression of *GAME* genes. A weaker *GEI* allele is proposed to be the main driver for selecting reduced SGA levels during tomato domestication. It is quite interesting to see the role of *GEI* in regulation of multiple *GAME* genes. However, I have major concerns for this manuscript.

Response: We appreciate Reviewer #2’s summary of our work and his/her point “It is quite interesting to see the role of *GEI* in regulation of multiple *GAME* genes”. We also appreciate reviewer #2’s primary concerns about our paper, which gave us food for thoughts to improve our work.

1. It lacks genetic evidence of *GEI* function. Deletion mutations, which can be readily developed using CRISPR/Cas in tomato, are absolutely required to provide direct evidence for the function of *GEI* as well as the functional difference between *GEI* and *GEI*⁷⁶. Most of the data in this manuscript, including DHS, histone modifications, and Hi-C, provide only correlative or indirect evidence for enhancer function.

Response: We agree with the reviewer that a genetic demonstration of *GEI* function would provide direct evidence supporting our conclusion. We are aware that this is a crucial point. In fact, we tried hard to generate *GEI* knockout tomato lines (*geI*), but this task proved extremely difficult because long-fragment deletion using CRISPR/Cas is quite challenging and to our knowledge this has only been reported once in tomato (Rodriguez-Leal *et al.*, 2017, *Cell*).

Nevertheless, we have persisted in such a strategy and have finally obtained three *GEI* knockout

tomato lines (*ge1-1*, *ge1-2*, and *ge1-3*) carrying long fragment deletion (See figure below; Figure 4a in our revised manuscript). Interestingly, the *ge1* lines displayed wrinkled surface that phenocopied the *ws* mutants and the transcript levels of *GAME* genes (*GAME1*, *GAME17* and *GAME18*) were significantly lower compared to that in WT. Consistent with the expression levels of *GAME* genes, the content of toxic SGAs including tomatidenol and tomatidine was significantly increased, while the accumulation of downstream SGAs compounds were decreased in the three *ge1* lines. We hope these new data provide the direct genetic evidence missing in our previous manuscript in supporting the function of *GE1* in regulation of the *GAME* gene cluster.

Generation of *ge1* knockout lines in MicroTom tomato by CRISPR/Cas9-mediated gene editing technology.

a, *ge1-1*, *ge1-2*, and *ge1-3* are three *GE1* knockout lines. Deletions (-) and insertions (+) indicated by numbers. Different gRNAs are presented by black arrowheads. b, Phenotypes of three *ge1* mutant fruits at 17 DPA (day-post-anthesis). Bar = 1 cm. c, RT-qPCR analysis of the relative expression of six *GAME* genes in three *ge1* lines and WT at 17 DPA. Data are presented as mean \pm SD (n=3). * $P < 0.05$ in Student's *t*-test. d, SGA profiles of *ge1* and WT fruits at 17 DPA. UPLC-MS was used for SGA content measurement. Data are presented as mean \pm SD (n=3). * $P < 0.05$ in Student's *t*-test.

2. The authors have ignored the extensive recent literature on the cis-regulatory function of DNA sequences associated with accessible chromatin regions (DHSs). Based on the literature, the authors should focus on the DHSs within *GEI*, rather than dividing *GEI* into three different segments based on information of H3K27ac, which has recently been disapproved as an enhancer mark in *Arabidopsis thaliana* (Yan *et al.*, 2019, *Nature Communications*).

Response: We thank the reviewer for this critical and constructive comment. We believe there is a misunderstanding regarding this issue due to a lack of clarity in our description of this part in the initial manuscript. Actually, we did use the DHSs data (please refer to upper panel of Figure 2a) as the most important feature to identify the enhancer *GEI*. Moreover, we divided *GEI* into three different segments based on the DHSs instead of the information of H3K27ac, since the three segments display different levels of chromatin accessibility, while the H3K27ac levels remain relatively stable in the three segments. Hope this clarifies the issue.

On the other hand, we agree with the reviewer that some studies (like Yan *et al.*, 2019, *Nat. Commun*) indicated that H3K27ac not a hallmark for enhancers in *Arabidopsis*. However, it is also important to mention that several recent literatures suggest that H3K27ac can be used as an enhancer marker in different species, such as in human (Ying *et al.*, 2023, *Nat. Commun*), rice (Joly-Lopez *et al.*, 2020, *Nat. Plants*), maize (Li *et al.*, 2019, *Nat. Commun*), and in tomato (Huang *et al.*, 2023, *Nat. Commun*). In our study, we identified *GEI* primarily based on DHSs data, while also considering histone modification data, such as H3K27ac, H3K27me3 and H3K4me3.

3. The Hi-C data is not well described and illustrated with details. I am not able to see the interaction between the *GEI* region and the individual *GAME* genes. In addition, the authors didn't explain why the T-DNA insertion only disrupts the contact on one side of *GEI*, and why the disruption from T-DNA would up-regulate (not down-regulate) *GAME1* and *GAME17*.

Response: We understand the reviewer comment regarding the figure presenting the Hi-C data and we provide now a new figure with improved resolution to clearly show the interaction between *GEI* region and the individual *GAME* genes (See figure below; Figure 3a-f in our revised manuscript). The solid black line represents the topologically associating domains (TADs) indicated by white arrows, and the red box represents the chromatin interaction regions indicated by green arrows. The difference in chromatin loops between WT and *ws* mutant was marked with green circles.

The interactions between *GEI* region and the three *GAME* cluster genes in WT (*GAME1*, *GAME17*, and *GAME18*) correspond to the two red boxes in the top of the green circles (See figure below; Figure 3c in our revised manuscript). Remarkably, these interactions disappeared in *ws* mutant

carrying the T-DNA insertion. Following the reviewer's suggestion, we have now added more detailed description of the Hi-C data in the figure legends.

Although we don't have a precise explanation regarding the reasons why the T-DNA insertion only disrupts the contact on one side of *GEI*, a possible explanation might be that the chromatin loops on the right side are those being directly regulated by *GEI*.

Regarding the up-regulation of *GAME1* and *GAME17*, it is reported that if the interaction of enhancer and target genes had been disrupted, the released enhancer RNA (eRNA) can retarget other promoter regions in the same contact domain, resulting in an increased expression of the adjacent genes, a mechanism known as the "Enhancer release and retargeting (ERR)" model (Oh *et al.*, 2021, *Nature*). In our study, we revealed that the loss of spatial interaction between *GEI* and GAME cluster resulted in an increase in the eRNA of *GEI*. *GEI* is located in the intergenic region with the two closest genes being *GAME1* and *GAME17*. In *ws* mutant, although the spatial interactions between *GEI* and GAME cluster had been disrupted, the increased *GEI* eRNA may preferentially target the genes closest to the enhancer, *i.e.*, *GAME1* and *GAME17*, according to the ERR model. *GEI* and *GAME18* are quite far away from each other (~52 kb), which might make it difficult for *GEI* eRNA to target *GAME18* when losing chromatin spatial interactions. Thus, the increased eRNA in the *ws* mutant is a likely cause of the rise in *GAME1* and *GAME17* expression, some similar findings have been reported (Field *et al.* 2008, *Science*; Zhu *et al.* 2015, *Plant Cell*; Zhao *et al.* 2022, *PNAS*).

It is worth pointing out that unlike the situation in the T-DNA insertion lines (*ws* mutant), the increase of *GEI* eRNA may not happen in *GEI* knockout lines due to the deletion of a long fragment within the *GEI* region and thus the expression of the three GAME cluster genes (*GAME1*, *GAME17*, and *GAME18*), that are the *GEI* targets, were all downregulated in *GEI* knockout lines (Figure 6c-f in our revised manuscript).

Chromatin interaction heatmaps of chromosome 7 in WT and *ws* fruits at 100 kb (a and d), 10 kb (b and e), or 5 kb (c and f) resolution. The solid black line represents the topologically associating domains (TADs), and the red box represents the chromatin interactions region. Solid black lines and red boxes are marked by white and green arrows, respectively. The difference of chromatin loops between WT and *ws* mutant is marked with green circles.

4. The authors briefly cited a recent publication (Zhao *et al.*, 2022, *PNAS*) of an enhancer in *Arabidopsis thaliana*, which similarly regulates multiple genes in a biosynthetic gene cluster, and used a very similar diagram in Figure 6 as Zhao *et al.*, (Figure 5). However, there is no discussion about the similarity and difference of these two enhancers.

Response: This is an excellent suggestion. As a classical enhancer working model, our model of GE1-mediated regulation on *GAME* gene cluster has similarity to the reported modulation by a super enhancer (SE) of thalianol gene cluster in *Arabidopsis* (Zhao *et al.*, 2022, *PNAS*), suggesting that enhancers may serve as key regulators in modulating the expression of metabolic gene clusters in different plants. However, *GE1* is a single enhancer, rather than a super-enhancer that consists of multiple enhancers as it is the case for SE in in *Arabidopsis*. We have now discussed the similarity and differences of these two enhancers in our revised manuscript.

References

- Field, B. & Osbourn, A. E. (2008). Metabolic Diversification-Independent Assembly of Operon-Like Gene Clusters in Different Plants. *Science* **320**, 543-547.
- Huang, Y., An, J., Sircar, S., *et al.* (2023). HSF1a modulates plant heat stress responses and alters the 3D chromatin organization of enhancer-promoter interactions. *Nat. Commun.* **14**, 469.

- Joly-Lopez, Z., Platts, A.E., Gulko, B., *et al.* (2020). An inferred fitness consequence map of the rice genome. *Nat. Plants* **6**, 119-130.
- Li, E., Liu, H., Huang, L., *et al.* (2019). Long-range interactions between proximal and distal regulatory regions in maize. *Nat. Commun.* **10**, 2633.
- Oh, S., Shao, J., Mitra, J., *et al.* (2021). Enhancer release and retargeting activates disease-susceptibility genes. *Nature* **595**, 735-740.
- Rodriguez-Leal, D., Lemmon, Z.H., Man, J., Bartlett, M.E., and Lippman, Z.B. (2017). Engineering Quantitative Trait Variation for Crop Improvement by Genome Editing. *Cell* **171**, 470-480 e478.
- Zhao, H., Yang, M., Bishop, J., *et al.* (2022). Identification and functional validation of super-enhancers in *Arabidopsis thaliana*. *PNAS* **119**, e2215328119.
- Zhu, B., Zhang, W.L., Zhang, T., *et al.* (2015) Genome-wide prediction and validation of intergenic enhancers in *Arabidopsis* using open chromatin signatures. *Plant Cell* **27**: 2415–2426
- Yan, W., *et al.* (2019). Dynamic control of enhancer activity drives stage-specific gene expression during flower morphogenesis. *Nat Commun* **10**, 1705.
- Ying, P., Chen, C., Lu, Z. *et al.* (2023) Genome-wide enhancer-gene regulatory maps link causal variants to target genes underlying human cancer risk. *Nat Commun* **14**, 5958.

REVIEWER COMMENTS

Reviewer #1 (Remarks to the Author):

The authors have responded to most of the reviewers' comments. However, it would be good to see tomatine data for more varieties in Figure 5. Only two accessions for each genotype are used to generate these graphs, and it is difficult to see that the tomatine content really varies between the GE1 and GE176 accessions from figure 5c. I see error bars for the GE176, but not for GE1. Could seeds be obtained for more accessions? I believe some of these accessions are publicly available.

Reviewer #2 (Remarks to the Author):

I am pleased to acknowledge the significant improvements that have been incorporated in response to my earlier comments. I believe that the revised manuscript is now well-prepared for publication.

Point-by-point responses to Reviewer #1 (Remarks to the Author):

The authors have responded to most of the reviewers' comments. However, it would be good to see tomatine data for more varieties in Figure 5. Only two accessions for each genotype are used to generate these graphs, and it is difficult to see that the tomatine content really varies between the *GE1* and *GE1*⁷⁶ accessions from figure 5c. I see error bars for the *GE1*⁷⁶, but not for *GE1*. Could seeds be obtained for more accessions? I believe some of these accessions are publicly available.

Response: We agree with the reviewer that it is important to see SGAs data for more varieties. Therefore, we made our effort to seek for the seeds from the first round of revision and obtained 19 varieties including 13 *GE1* (TS2, TS3, TS9, TS15, TS60, TS96, TS118, TS166, TS171, TS185, TS204, TS256, and TS439) and 6 *GE1*⁷⁶ (TS22, TS39, TS117, TS156, TS413, and LA2093) accessions from the Institute of Vegetable and Flower Research, Chinese Academy of Agricultural Sciences (IVF-CAAS). The data showed that both the *GAME* expression levels and SGAs contents are relatively higher in the six *GE1*⁷⁶ accessions compared to *GE1* accessions (See Figure below, Fig. 5d and 5e in our revised manuscript). Regarding Fig. 5c, we have now provided a new Figure with error bars for both *GE1* and *GE1*⁷⁶ accessions. Based on the publicly available data (Fig. 5c) and our newly provided data (Fig. 5d and 5e), we can clearly see the varies of SGA content between the *GE1* and *GE1*⁷⁶ accessions.

Upper panel, RT-qPCR analysis showing the expression levels of six *GAME* genes clustered on chr07 in 19 tomato varieties. Nineteen varieties with 13 contain *GE1* accessions (TS2, TS3, TS9, TS15, TS60, TS96, TS118, TS166, TS171, TS185, TS204, TS256, and TS439) and other 6 contain *GE1*⁷⁶ accessions (TS22, TS39, TS117, TS156, TS413, and LA2093). Dots indicate individual tomato accession ($n_{GE1} = 13$; $n_{GE1}76 = 6$). * $P < 0.05$ in Student's *t*-test.

Lower panel, SGAs profiles of nineteen varieties of *GE1*- and *GE1*⁷⁶-containing accessions. Dots indicate individual tomato accession ($n_{GE1} = 13$; $n_{GE1}76 = 6$). * $P < 0.05$ in Student's *t*-test.

Point-by-point responses to Reviewer #2 (Remarks to the Author):

I am pleased to acknowledge the significant improvements that have been incorporated in response to my earlier comments. I believe that the revised manuscript is now well-prepared for publication.

Response: We thank the reviewer for the valuable and instructive comments to improve our manuscript.

REVIEWERS' COMMENTS

Reviewer #1 (Remarks to the Author):

The authors have responded to my review of the manuscript and I feel it is now ready for publication.